# Spectral bias and task-model alignment explain generalization in kernel regression and infinitely wide neural networks

Abdulkadir Canatar [1,2], Blake Bordelon[2,3] & Cengiz Pehlevan [2,3 ✉]

A theoretical understanding of generalization remains an open problem for many machine learning models, including deep networks where overparameterization leads to better performance, contradicting the conventional wisdom from classical statistics. Here, we investigate generalization error for kernel regression, which, besides being a popular machine learning method, also describes certain infinitely overparameterized neural networks. We use techniques from statistical mechanics to derive an analytical expression for generalization error applicable to any kernel and data distribution. We present applications of our theory to real and synthetic datasets, and for many kernels including those that arise from training deep networks in the infinite-width limit. We elucidate an inductive bias of kernel regression to explain data with simple functions, characterize whether a kernel is compatible with a learning task, and show that more data may impair generalization when noisy or not expressible by the kernel, leading to non-monotonic learning curves with possibly many peaks.

[1] Department of Physics, Harvard University, Cambridge, MA, USA. [2] Center for Brain Science, Harvard University, Cambridge, MA, USA. [3] John A. Paulson School of Engineering and Applied Sciences, Harvard University, Cambridge, MA, USA. ✉email: cpehlevan@seas.harvard.edu

Learning machines aim to find statistical patterns in data that generalize to previously unseen samples[1]. How well they perform in doing so depends on factors such as the size and the nature of the training data set, the complexity of the learning task, and the inductive bias of the learning machine. Identifying precisely how these factors contribute to the generalization performance has been a theoretical challenge. In particular, a definitive theory should be able to predict generalization performance on real data. Existing theories fall short of this goal, often providing impractical bounds and inaccurate estimates[2,3].

The need for a new theory of generalization is exacerbated by recent developments in deep learning[4]. Experience in the field suggests that larger models perform better[5–7], encouraging training of larger and larger networks with state-of-the-art architectures reaching hundreds of billions of parameters[7]. These networks work in an overparameterized regime[3,5] with much more parameters than training samples, and are highly expressive to a level that they can even fit random noise[2]. Yet, they generalize well, contradicting the conventional wisdom from classical statistical learning theory[1,3,8] according to which overparameterization should lead to overfitting and worse generalization. It must be that overparameterized networks have inductive biases that suit the learning task. Therefore, it is crucial for a theory of generalization to elucidate such biases.

While addressing the full complexity of deep learning is as of now beyond the reach of theoretical study, a tractable, yet practically relevant limit was established by recent work pointing to a correspondence between training deep networks and performing regression with various rotation invariant kernels. In the limit where the width of a network is taken to infinity (network is thus overparameterized), neural network training with a certain random initialization scheme can be described by ridgeless kernel regression with the Neural Network Gaussian Process kernel (NNGPK) if only the last layer is trained[9–12], or the Neural Tangent Kernel (NTK) if all the layers are trained[13]. Consequently, studying the inductive biases of kernels arising from the infinite-width limit should give insight to the success of overparameterized neural networks. Indeed, key generalization phenomena in deep learning also occur in kernel methods, and it has been argued that understanding generalization in kernel methods is necessary for understanding generalization in deep learning[14].

Motivated by these connections to deep networks and also by its wide use, in this paper, we present a theory of generalization in kernel regression[15–19]. Our theory is generally applicable to any kernel and contains the infinite-width limit of deep networks as a special case. Most importantly, our theory is applicable to real datasets.

We describe typical generalization performance of kernel regression shedding light onto practical uses of the algorithm, in contrast to the worst case bounds of statistical learning theory[8,18,20–22]. In the past, statistical mechanics provided a useful theoretical framework for such typical-case analyses for various algorithms[23–32]. Here, using the replica method of statistical mechanics[33], we derive an analytical expression for the typical generalization error of kernel regression as a function of (1) the number of training samples, (2) the eigenvalues and eigenfunctions of the kernel, which define the inductive bias of kernel regression, and (3) the alignment of the target function with the kernel's eigenfunctions, which provides a notion of how compatible the kernel is for the task. We test our theory on various real datasets and kernels. Our analytical generalization error predictions fit experiments remarkably well.

Our theory sheds light onto the various generalization phenomena. We elucidate a strong inductive bias: as the size of the training set grows, kernel regression fits successively higher spectral modes of the target function, where the spectrum is defined by solving an eigenfunction problem[19,34–36]. Consequently, our theory can predict which kernels or neural architectures are well suited to a given task by studying the alignment of top kernel eigenfunctions with the target function for the task. Target functions that place most power in the top kernel eigenfunctions can be estimated accurately at small sample sizes, leading to good generalization. Finally, when the data labels are noisy or the target function has components not expressible by the kernel, we observe that generalization error can exhibit non-monotonic behavior as a function of the number of samples, contrary to the common intuition that more data should lead to smaller error. This non-monotonic behavior is reminiscent of the recently described "double-descent" phenomenon[3,5,37,38], where generalization error is non-monotonic as a function of model complexity in many modern machine learning models. We show that the non-monotonicity can be mitigated by increasing the implicit or explicit regularization.

To understand these phenomena better, we present a detailed analytical study of the application of our theory to rotation invariant kernels, motivated by their wide use and relevance for deep learning. Besides NNGPK and NTK, this class includes many other popular kernels such as the Gaussian, Exponential and Matern kernels[39,40]. When the data generating distribution is also spherically symmetric, our theory is amenable to further analytical treatment. Our analyses provide a mechanistic understanding of the inductive bias of kernel regression and the possible non-monotonic behavior of learning curves.

## Results

**Generalization error of kernel regression from statistical mechanics.** Kernel regression is a supervised learning problem where one estimates a function from a number of observations. For our setup, let $\mathcal{D} = \{\mathbf{x}^\mu, y^\mu\}_{\mu=1}^P$ be a sample of $P$ observations drawn from a probability distribution on $\mathcal{X} \times \mathbb{R}$, and $\mathcal{X} \subseteq \mathbb{R}^D$. The inputs $\mathbf{x}^\mu$ are drawn from a distribution $p(\mathbf{x})$, and the labels $y^\mu$ are assumed to be generated by a noisy target $y^\mu = \bar{f}(\mathbf{x}^\mu) + \epsilon^\mu$, where $\bar{f}$ is square integrable with respect to $p(\mathbf{x})$, and $\epsilon^\mu$ represents zero-mean additive noise with covariance $\langle \epsilon^\mu \epsilon^\nu \rangle = \delta_{\mu\nu}\sigma^2$. The kernel regression problem is

$$f^* = \arg\min_{f \in \mathcal{H}} \frac{1}{2\lambda} \sum_{\mu=1}^P (f(\mathbf{x}^\mu) - y^\mu)^2 + \frac{1}{2}\langle f, f \rangle_{\mathcal{H}}, \quad (1)$$

where $\lambda$ is the "ridge" parameter, $\mathcal{H}$ is a Reproducing Kernel Hilbert Space (RKHS) uniquely determined by its reproducing kernel $K(\mathbf{x}, \mathbf{x}')$ and the input distribution $p(\mathbf{x})$[41], and $\langle \cdot, \cdot \rangle_{\mathcal{H}}$ is the RKHS inner product. The Hilbert norm penalty controls the complexity of $f$. The $\lambda \to 0$ limit is referred to as the kernel interpolation limit, where the dataset is exactly fit: $f^* = \arg\min_{f \in \mathcal{H}} \langle f, f \rangle_{\mathcal{H}}$, s.t. $f(\mathbf{x}^\mu) = y^\mu, \mu = 1, \dots P$. We emphasize that in our setting the target function does not have to be in the RKHS.

Our goal is to calculate generalization error, i.e. mean squared error between the estimator, $f^*$, and the ground-truth (target) $\bar{f}(\mathbf{x})$ averaged over the data distribution and datasets:

$$E_g = \left\langle \int d\mathbf{x}\, p(\mathbf{x}) \left( f^*(\mathbf{x}) - \bar{f}(\mathbf{x}) \right)^2 \right\rangle_{\mathcal{D}}. \quad (2)$$

$E_g$ measures, on average, how well the function learned agrees with the target on previously unseen (and seen) data sampled from the same distribution.

This problem can be analyzed using the replica method from statistical physics of disordered systems[33], treating the training set as a quenched disorder. Our calculation is outlined in the Methods and further detailed in the Supplementary Information. Here we present our main results.

Our results rely on the Mercer decomposition of the kernel in terms of orthogonal eigenfunctions $\{\phi_\rho\}$,

$$\int d\mathbf{x}' p(\mathbf{x}') K(\mathbf{x}, \mathbf{x}') \phi_\rho(\mathbf{x}') = \eta_\rho \phi_\rho(\mathbf{x}), \qquad \rho = 1, \ldots, N, \quad (3)$$

which form a complete basis for the RKHS, and eigenvalues $\{\eta_\rho\}$. $N$ is typically infinite. For ease of presentation, we assume that all eigenvalues are strictly greater than zero. In Supplementary Notes 1 and 2, we fully address the case with zero eigenvalues. Working with the orthogonal basis set $\psi_\rho(\mathbf{x}) \equiv \sqrt{\eta_\rho} \phi_\rho(\mathbf{x})$, also called a feature map, we introduce coefficients $\{\overline{w}_\rho\}$ and $\{w_\rho^*\}$ that represent the target and the estimator respectively: $\bar{f}(\mathbf{x}) = \sum_\rho \overline{w}_\rho \psi_\rho(\mathbf{x})$, and $f^*(\mathbf{x}) = \sum_\rho w_\rho^* \psi_\rho(\mathbf{x})$.

With this setup, we calculate the generalization error of kernel regression for any kernel and data distribution to be (Methods and Supplementary Note 2):

$$E_g = \frac{1}{1 - \gamma} \sum_\rho \frac{\eta_\rho}{\left(\kappa + P\eta_\rho\right)^2} \left(\kappa^2 \overline{w}_\rho^2 + \sigma^2 P\eta_\rho\right),$$

$$\kappa = \lambda + \sum_\rho \frac{\kappa \eta_\rho}{\kappa + P\eta_\rho}, \quad \gamma = \sum_\rho \frac{P\eta_\rho^2}{(\kappa + P\eta_\rho)^2}. \quad (4)$$

We note that the generalization error is the sum of a $\sigma$-independent term and a $\sigma$-dependent term, the latter of which fully captures the effect of noise on generalization error.

Formally, this equation describes the typical behavior of kernel regression in a thermodynamic limit that involves taking $P$ to infinity. In this limit, variations in kernel regression's performance due to the differences in how the training set is formed, which is assumed to be a stochastic process, become negligible. The precise nature of the limit depends on the kernel and the data distribution. In this work, we consider two different analytically solvable cases and identify natural scalings of $N$ and $D$ with $P$, which in turn govern how the kernel eigenvalues $\eta_\rho$ scale inversely with $P$. We further give the infinite-$P$ limits of Eq. (4) explicitly for these cases. In practice, however, we find that our generalization error formula describes average learning curves very well for finite $P$ for even as low as a few samples. We observe that the variance in learning curves due to stochastic sampling of the training set is significant for low $P$, but decays with increasing $P$ as expected.

We will demonstrate various generalization phenomena that arises from Eq. (4) through simulations and analytical study. One immediate observation is the spectral bias: faster rates of convergence of the error along eigenfunctions corresponding to higher eigenvalues in the noise-free ($\sigma^2 = 0$) limit. The generalization error can be decomposed into a sum of modal errors $E_g = \sum_\rho \eta_\rho \overline{w}_\rho^2 E_\rho$, where each normalized mode error $E_\rho = \frac{1}{\overline{w}_\rho^2} \langle (w_\rho^* - \overline{w}_\rho)^2 \rangle_{\mathcal{D}}$ represents the contribution of the mode error due to estimation of the coefficient for eigenfunction $\psi_\rho$ (Methods). The normalized mode errors are ordered according to their eigenvalues for all $P$ (Methods)

$$\eta_\rho > \eta_{\rho'} \implies E_\rho < E_{\rho'}, \quad (5)$$

which implies that modes $\rho$ with large eigenvalues $\eta_\rho$ are learned more rapidly as $P$ increases than modes with small eigenvalues.

An important implication of this result is that target functions acting on the same data distribution with higher cumulative power distributions $C(\rho)$, defined as the proportion of target power in the first $\rho$ modes

$$C(\rho) = \frac{\sum_{\rho' \le \rho} \eta_{\rho'} \overline{w}_{\rho'}^2}{\sum_{\rho'} \eta_{\rho'} \overline{w}_{\rho'}^2}, \quad (6)$$

for all $\rho \ge 1$ will have lower generalization error normalized by total target power, $E_g(P)/E_g(0)$, for all $P$ (Methods). Therefore, $C(\rho)$ provides a measure of the compatibility between the kernel and the target, which we name task-model alignment.

We further note that the target function enters normalized generalization error only through combinations $C(\rho) - C(\rho - 1) = \eta_\rho \overline{w}_\rho^2 / \sum_\rho \eta_\rho \overline{w}_\rho^2$. Hence, the kernel eigenvalues, the cumulative power distribution, and the noise parameter completely specify the normalized generalization error. Spectral bias, task-model alignment and noise explain generalization in kernel regression.

Generalization error can exhibit non-monotonicity which can be understood through the bias and variance decomposition[38,42,43], $E_g = B + V$, where $B = \int d\mathbf{x} p(\mathbf{x}) \left( \langle f^*(\mathbf{x}) \rangle_{\mathcal{D}} - \bar{f}(\mathbf{x}) \right)^2$ and $V = \left\langle \int d\mathbf{x} p(\mathbf{x}) (f^*(\mathbf{x}) - \langle f^*(\mathbf{x}) \rangle_{\mathcal{D}})^2 \right\rangle_{\mathcal{D}}$. We found that the average estimator is given by $\langle f^*(\mathbf{x}; P) \rangle_{\mathcal{D}} = \sum_\rho \frac{P\eta_\rho}{P\eta_\rho + \kappa} \overline{w}_\rho \psi_\rho(\mathbf{x})$, which monotonically approaches to the target function as $P$ increases, giving rise to a monotonically decreasing bias (Supplementary Note 2). However, the variance term arising from the variance of the estimator over possible sampled datasets $\mathcal{D}$ is potentially non-monotonic as the dataset size increases. Therefore, the total generalization error can exhibit local maxima.

**Applications to real datasets**. Next, we evaluate our theory on realistic datasets and show that it predicts kernel regression learning curves with remarkable accuracy. We further elucidate various heuristic generalization principles.

To apply our theory, we numerically solve the eigenvalue problem Eq. (3) on the dataset (Methods) and obtain the necessary eigenvalues and eigenfunctions. When solved on a finite dataset, Eq. (3) is an uncentered kernel PCA problem (Methods). We use these eigenfunctions (or eigenvectors for finite data) to express our target function, and the resulting coefficients and kernel eigenvalues to evaluate the generalization error.

In our first experiment, we test our theory using a 2-layer NTK[10,13] on two different tasks: discriminating between 0 and 1 s, and between 8 and 9 s from MNIST dataset[44]. We formulate each of these tasks as a kernel regression problem by considering a vector target function which takes in digits and outputs one-hot labels. Our kernel regression theory can be applied separately to each element of the target function vector (Methods), and a generalization error can be calculated by adding the error due to each vector component.

We can visualize the complexity of the two tasks by plotting the projection of the data along the top two kernel principal components (Fig. 1a, b). The projection for 0–1 digits appears highly separable compared to 8–9s, and thus simpler to learn to discriminate. Indeed, the generalization error for the 0–1 discrimination task falls more rapidly than the error for the 8–9 task (Fig. 1c). Our theory is in remarkable agreement with experiments. Why is 0–1 discrimination easier for this kernel? Fig. 1d shows that the eigenvalues of the NTK evaluated on the data are very similar for both datasets. To quantify the compatibility of the kernel with the tasks, we measure the cumulative power distribution $C(\rho)$. Even though in this case the data distributions are different, $C(\rho)$ is still informative. Figure 1e illustrates that $C(\rho)$ rises more rapidly for the easier 0–1 task and more slowly for the 8–9 task, providing a heuristic explanation of why it requires a greater number of samples to learn.

We next test our theory for Gaussian RBF kernel on the MNIST[44] and CIFAR[45] datasets. Figure 2a shows excellent agreement between our theory and experiments for both. Figure 2b shows that the eigenvalues of the Gaussian RBF kernel

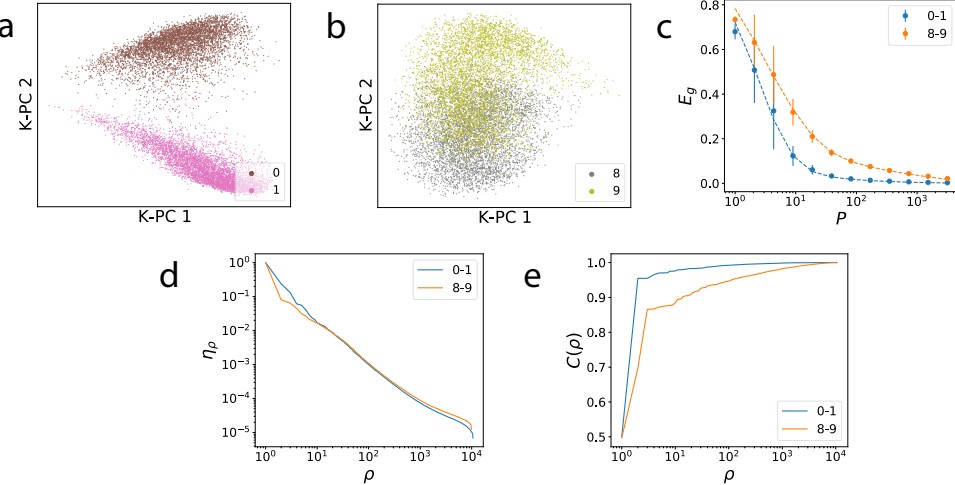

**Fig. 1 Effect of task-model alignment on the generalization of kernel regression. a, b** Projections of digits from MNIST along the top two (uncentered) kernel principal components of 2-layer NTK for 0s vs. 1s and 8s vs. 9s, respectively. **c** Learning curves for both tasks. The theoretical learning curves (Eq. (4), dashed lines) show strong agreement with experiment (dots). **d** The kernel eigenspectra for the respective datasets. **e** The cumulative power distributions $C(\rho)$. Error bars show the standard deviation over 50 trials.

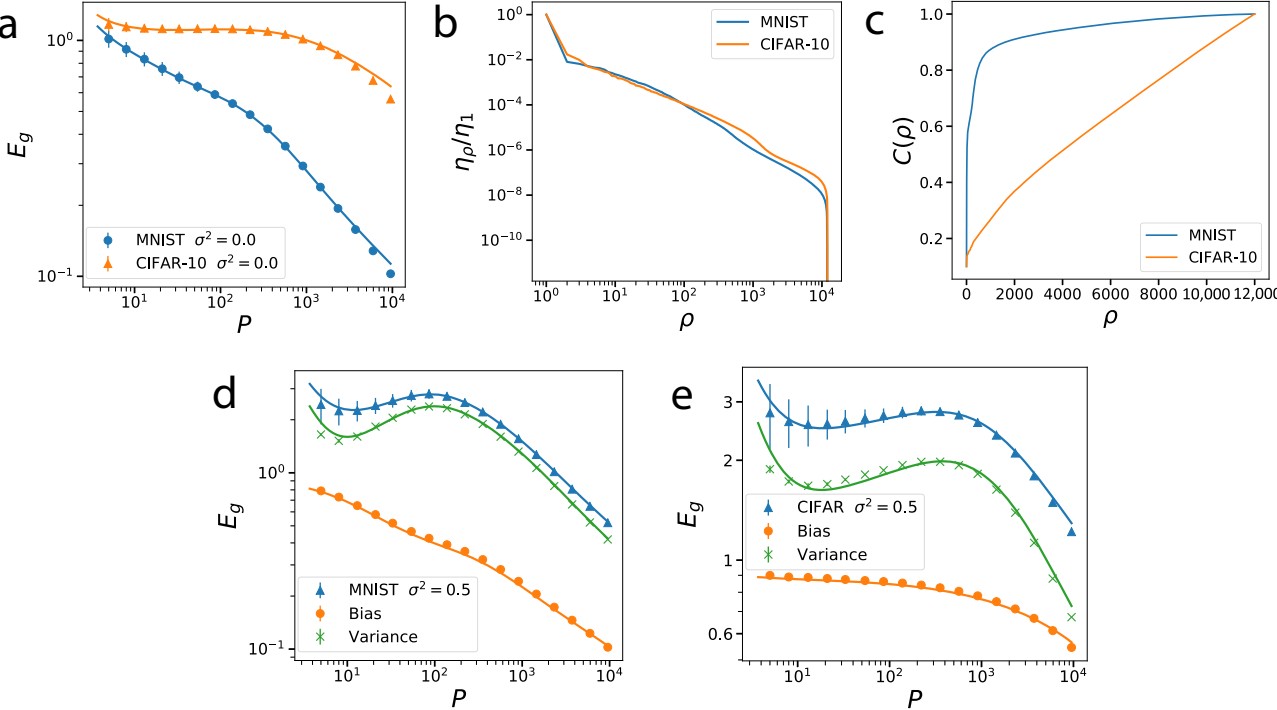

**Fig. 2 Gaussian RBF kernel regression on MNIST and CIFAR-10 datasets.** Kernel is $K(\mathbf{x}, \mathbf{x}') = e^{-\frac{1}{2D\omega^2}||\mathbf{x}-\mathbf{x}'||^2}$ with kernel bandwidth $\omega = 0.1$, ridge parameter $\lambda = 0.01$ and $D$ being the size of images. **a** Generalization error $E_g(p)$ when $\sigma^2 = 0$: Solid lines are theory (Eq. (4)), dots are experiments. **b** Kernel eigenvalues and **c** cumulative powers $C(\rho)$ for MNIST and CIFAR-10. **d, e** Generalization error when $\sigma^2 = 0.5$ with its bias-variance decomposition for MNIST and CIFAR-10 datasets, respectively. Solid lines are theory, markers are experiments. Error bars represent standard deviation over 160 trials. Bias and variance are obtained by calculating the mean and variance of the estimator over 150 trials, respectively.

evaluated on data are similar for MNIST and CIFAR-10. The cumulative powers $C(\rho)$ (Fig. 2c), however, are very different. Placing more power in the first few modes makes learning faster. When the labels have nonzero noise $\sigma^2 > 0$ (Fig. 2d, e), generalization error is non-monotonic with a peak, a feature that has been named "double-descent"[3,37]. By decomposing $E_g$ into the bias and the variance of the estimator, we see that the non-monotonicity is caused solely by the variance (Fig. 2d, e). Similar observations about variance were made in different contexts before[38,42,46].

These experiments and discussion in the previous section provide illustrations of the three main heuristics about how dataset, kernel, target function, and noise interact to produce generalization in kernel regression:

(1) Spectral Bias: Kernel eigenfunctions $\phi_\rho$ with large eigenvalues $\eta_\rho$ can be estimated with kernel regression using a smaller number of samples.

(2) Task-Model Alignment: Target functions with most of their power in top kernel eigenfunctions can be estimated

efficiently and are compatible with the chosen kernel. We introduce cumulative power distribution, $C(\rho)$, as defined in Eq. (6), as a measure of this alignment.

(3) Non-monotonicity: Generalization error may be non-monotonic with dataset size in the presence of noise (as in Fig. 2), or when the target function is not expressible by the kernel (not in the RKHS). We provide a discussion of and examples for the latter kind in Supplementary Notes 3 and 4. We show that modes of the target function corresponding to zero eigenvalues of the kernel act effectively as noise on the learning problem.

To explore these phenomena further and understand their causes, we study several simplified models where the kernel eigenvalue problem and generalization error equations can be solved analytically.

**Double-descent phase transition in a band-limited RKHS.** An explicitly solvable and instructive case is the white band-limited RKHS with $N$ equal nonzero eigenvalues, a special case of which is linear regression. Later, we will observe that the mathematical description of rotation invariant kernels on isotropic distributions reduces to this simple model in each learning stage.

In this model, the kernel eigenvalues are equal $\eta_\rho = \frac{1}{N}$ for a finite number of modes $\rho = 1, ..., N$ and truncate thereafter: $\eta_\rho = 0$ for $\rho > N$. Similarly, the target power $\overline{w}_\rho^2$ truncates after $N$ modes and satisfies the normalization condition $\sum_{\rho=1}^{N} \overline{w}_\rho^2 = N$. In Supplementary Note 3, we relax these constraints and discuss their implications. Linear regression (or linear perceptron) with isotropic data is a special case when $D = N$, $\phi_\rho(\mathbf{x}) = x_\rho$, and $\langle x_\rho x_{\rho'} \rangle_{\mathbf{x} \sim p(\mathbf{x})} = \delta_{\rho\rho'}$[25].

We study this model in the thermodynamic limit. We find that the natural scaling is to take $P \to \infty$ and $N \to \infty$ with $\alpha = P/N \sim \mathcal{O}(1)$, and $D \sim O(1)$ (or $D = N \sim \mathcal{O}(P)$ in the linear regression case), leading to the generalization error:

$$E_g(\alpha, \lambda, \sigma^2) = \frac{\kappa_\lambda(\alpha)^2 + \sigma^2 \alpha}{(\kappa_\lambda(\alpha) + \alpha)^2 - \alpha},$$

$$\kappa_\lambda(\alpha) = \frac{1}{2}\left[(1 + \lambda - \alpha) + \sqrt{(1 + \lambda - \alpha)^2 + 4\lambda\alpha}\right]. \tag{7}$$

Note that this result is independent of the teacher weights as long as they are properly normalized. The function $\kappa_\lambda(\alpha)$ appears in many contexts relevant to random matrix theory, as it is related to the resolvent, or Stieltjes transform, of a random Wishart matrix[47,48] (Supplementary Note 3). This simple model shows interesting behavior, elucidating the role of regularization and under- vs. over-parameterization in learning machines.

First we consider the interpolation limit ($\lambda = 0$, Fig. 3a). The generalization error simplifies to $E_g = (1 - \alpha)\Theta(1 - \alpha) + \frac{\sigma^2}{1-\alpha}[\alpha\Theta(1 - \alpha) - \Theta(\alpha - 1)]$. There is a first order phase transition at $\alpha_c = 1$, when the number of samples $P$ is equal to the number of nonzero modes $N$ and therefore to the number of parameters, $\{\overline{w}_\rho\}$, that define the target function. The phase transition is signaled by the non-analytic behavior of $E_g$ and verifiable by computing the first-derivative of free energy (Supplementary Note 3). When $\sigma = 0$, $E_g$ linearly falls with more data and at the critical point generalization error goes to zero. With noise present, the behavior at the critical point changes drastically, and there is a singular peak in the generalization error due to the noise term of the generalization error (Fig. 3a). At this point the kernel machine is (over-)fitting exactly all data points, including noise. Then, as number of samples increase beyond the number of parameters ($\alpha > 1$), the machine is able to average over noise and the generalization error falls with asymptotic behavior $E_g \sim \sigma^2/\alpha$.

Our results are consistent with those previously obtained for the linear perceptron with a noisy target[25,49], which is a special case of kernel regression with a white band-limited spectrum.

When $\lambda > 0$ and $\sigma = 0$, $E_g$ decreases monotonically with $\alpha$ and is asymptotic to $E_g \sim \lambda^2/\alpha^2$ (Fig. 3b). A sharp change at $\alpha = 1$ is visible for small $\lambda$, reminiscent of the phase transition at $\lambda = 0$. When $\sigma > 0$ is sufficiently large compared to $\lambda$, non-monotonicity is again present, giving maximum generalization error at $\alpha \approx 1 + \lambda$ (Fig. 3c), with an asymptotic fall $E_g \sim \frac{\sigma^2}{\alpha}$.

We find that $E_g(\alpha)$ is non-monotonic when the noise level in target satisfies the following inequality (Fig. 3d and Supplementary Note 3):

$$\sigma^2 > \begin{cases} g(\lambda) & \lambda < 1 \\ 2\lambda + 1 & \lambda \geq 1 \end{cases}, \tag{8}$$

where $g(\lambda) = 3\lambda[3\lambda + 2 - 2\sqrt{1 + \lambda}\sqrt{9\lambda + 1}\cos\theta(\lambda)]$, and $\theta(\lambda) = \frac{1}{3}(\pi + \tan^{-1}\frac{8\sqrt{\lambda}}{9\lambda(3\lambda+2)-1})$. Although there is no strict phase transition (in the sense of non-analytic free energy) except at $\lambda = 0$, Eq. (8) defines a phase boundary separating the monotonic and non-monotonic learning curve regions for a given regularization parameter and noise. For a given $\lambda$, double-descent occurs for sufficiently high $\sigma^2$. In the non-monotonic region, there is a single local maximum when $\sigma^2 > 2\lambda + 1$, otherwise a local minima followed by a local maxima (we call only this kind of peak as the double-descent peak).

Based on this explicit formula, one could choose a large enough $\lambda$ to mitigate the peak to avoid overfitting for a given noise level (Fig. 3d). However, larger $\lambda$ may imply slower learning (see Fig. 3b and Supplementary Note 3) requiring more training samples to achieve the same generalization error. By inspecting the derivative $\frac{\partial E_g}{\partial \lambda} = 0$, we find that $\lambda^* = \sigma^2$ (yellow dashed line in Fig. 3d) is the optimal choice for ridge parameter, minimizing $E_g(\alpha)$ for a given $\sigma^2$ at all $\alpha$ (Fig. 3c). For $\lambda > \lambda^*$ the noise-free error term increases from the optimum whereas $\lambda < \lambda^*$ gives a larger noise term. Our result agrees with a similar observation regarding the existence of an optimal ridge parameter in linear regression[46].

Further insight to the phase transition can be gained by looking at the bias and the variance of the estimator[38,42,43]. The average estimator learned by kernel regression linearly approaches the target function as $\alpha \to 1$ (Supplementary Note 2): $\langle f^*(\mathbf{x}) \rangle_\mathcal{D} = \min\{\alpha, 1\}\overline{f}(\mathbf{x})$ (Fig. 4a), which indicates that the bias ($B$) and variance ($V$) contributions to generalization error have the forms $B = \max\{0, 1 - \alpha\}^2$, $V = \alpha(1 - \alpha)\Theta(1 - \alpha) + \frac{\sigma^2}{1-\alpha}[\alpha\Theta(1 - \alpha) - \Theta(\alpha - 1)]$. In the absence of noise, $\sigma = 0$, variance is initially low at small $\alpha$, reaches its maximum at $\alpha = 1/2$ and then decreases as $\alpha \to 1$ as the learned function concentrates around $\overline{f}$ (Fig. 4b). When there is noise, the phase transition at $\alpha = 1$ arises from the divergence in the variance $V$ of the learned estimator (Fig. 4c).

**Multiple learning episodes and descents: rotation invariant kernels and measures.** Next, we study kernel regression on high-dimensional spheres focusing on rotation invariant kernels, which satisfy $K(\mathbf{Ox}, \mathbf{Ox}') = K(\mathbf{x}, \mathbf{x}')$, where $\mathbf{O}$ is an arbitrary orthogonal matrix. This broad class of kernels includes widely used radial basis function kernels $K(\mathbf{x}, \mathbf{x}') = K(||\mathbf{x} - \mathbf{x}'||)$ (Gaussian, Laplace, Matern, rational quadratic, thin plate splines, etc) and dot product kernels $K(\mathbf{x}, \mathbf{x}') = K(\mathbf{x} \cdot \mathbf{x}')$ (polynomial kernels, NNGPK and NTK)[10,39,40].

When the data distribution is spherically isotropic $p(\mathbf{x}) = p(||\mathbf{x}||)$, we can separate Mercer eigenfunctions for rotation invariant kernels into radial and angular parts. The spherical parts depend on the

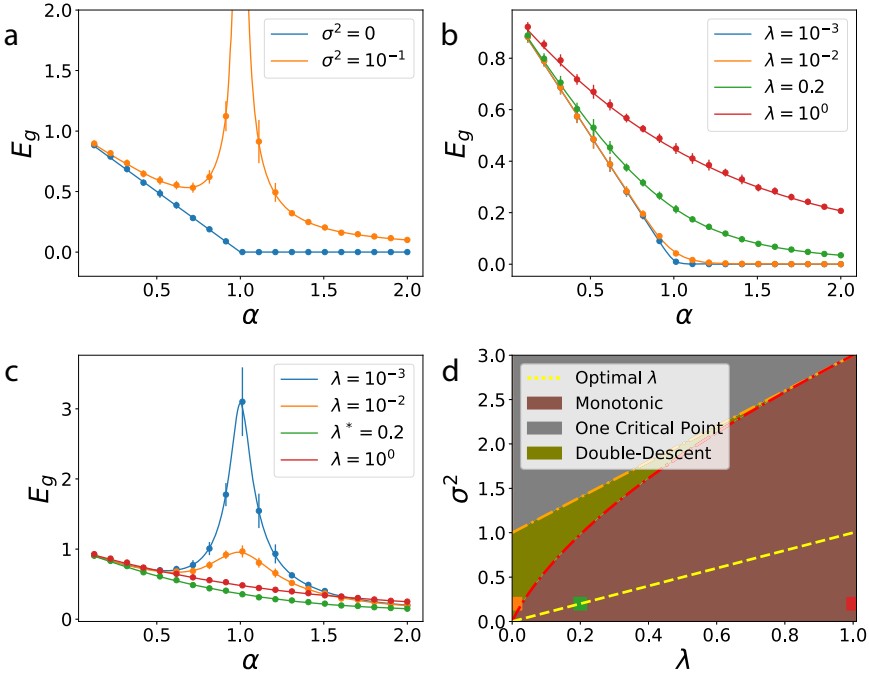

**Fig. 3 Learning curves and double-descent phase diagram for kernels with white band-limited spectra.** We simulated $N = 800$ dimensional uncorrelated Gaussian features $\phi(\mathbf{x}) = \mathbf{x} \sim \mathcal{N}(0, \mathbf{I})$ and estimated a linear function $\bar{f}(\mathbf{x}) = \boldsymbol{\beta}^\top \mathbf{x}$ with $\|\boldsymbol{\beta}\|^2 = N$. Error bars describe the standard deviation over 15 trials. Solid lines are theory (Eq. (7)), dots are experiments. **a** When $\lambda = 0$ and $\sigma^2 = 0$, $E_g$ linearly decreases with $\alpha$ and when $\sigma^2 > 0$ it diverges as $\alpha \to 1$. **b** When $\sigma^2 = 0$, explicit regularization $\lambda$ always leads to slower decay in $E_g$. **c** For nonzero noise $\sigma^2 > 0$, there is an optimal regularization $\lambda^* = \sigma^2$ which gives the best generalization performance. **d** Double-descent phase diagram where the colored squares correspond to the curves with same color in **c**. Optimal regularization ($\lambda^* = \sigma^2$) curve is shown in yellow dashed line which does not intersect the double-descent region above the curve defined by $g(\lambda)$ (Eq. (8)).

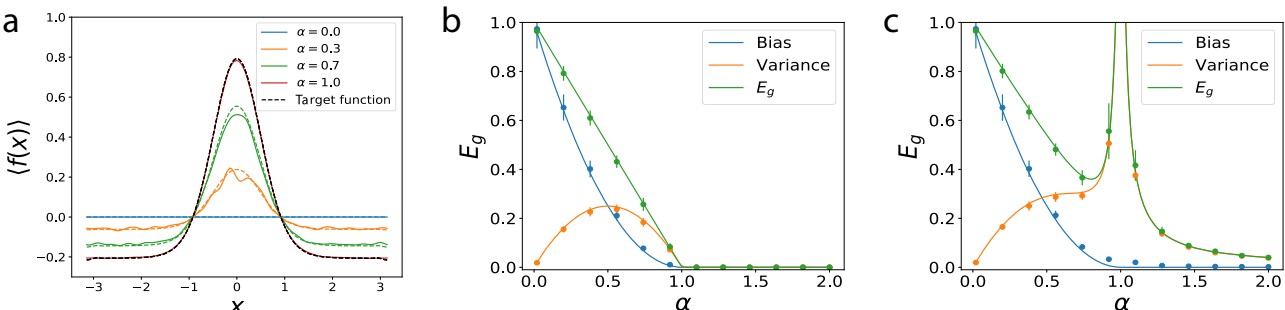

**Fig. 4 Bias-variance decomposition of generalization error. a** Average estimator for kernel regression with $K(x, x') = \sum_{k=1}^{N} \cos(k(x - x'))$ on target function $\bar{f}(x) = e^{4(\cos x - 1)}$ with mean subtracted for different values of $\alpha = P/N$ when $\lambda = \sigma^2 = 0$. Estimator linearly approaches to the target function and estimates it perfectly when $\alpha = 1$. Dashed lines are theory. **b** With the same setting in Fig. 3, when $\lambda = 0$ and $\sigma^2 = 0$, the bias is a monotonically decreasing function of $\alpha$ while variance has a peak at $\alpha = 1/2$ yet it does not diverge. **c** When $\lambda = 0$ and $\sigma^2 = 0.2$, we observe that the divergence of $E_g$ is only due to the diverging variance of the estimator. In **b**, **c**, solid lines are theory, dots are experiments. Error bars represent the standard deviation over 15 trials.

radial distances of the data points $\|\mathbf{x}\|$, whereas the angular components admit a decomposition in terms of spherical harmonics of the unit vectors $Y_{km}(\hat{x})$, where $k$ is the degree and $m$ is the order of the harmonic[50]. A review of the basic properties of spherical harmonics are provided in the Supplementary Note 6. Utilizing this spherical symmetry, we obtain the following Mercer decomposition $K(\mathbf{x}, \mathbf{x}') = \sum_{zkm} \eta_{z,k} R_{z,k}(\|\mathbf{x}\|) R_{z,k}(\|\mathbf{x}'\|) Y_{km}(\hat{\mathbf{x}}) Y_{km}(\hat{\mathbf{x}}')$. Since the eigenvalues are independent of the spherical harmonic order $m$, the minimal degeneracy of the RKHS spectrum is the number of degree $k$ harmonics: in the limit $D \to \infty$ given by $\sim D^k/k!$[51,52]. However, the degeneracy can be even larger if there are different $(z, k)$ indices with the same eigenvalue. For notational convenience, we denote degenerate eigenvalues as $\eta_K$ ($K \in \mathbb{Z}^+$) and corresponding eigenfunctions as $\phi_{K,\rho}$ where $\rho \in \mathbb{Z}^+$ indexes the degenerate indices.

After finding the eigenvalues of a kernel on the basis $\phi_{K,\rho}$, one can evaluate Eq. (4) to predict the generalization error of the kernel machine.

We focus on the case where the degeneracy of $\eta_K$ is $N(D, K) \sim \mathcal{O}_D(D^K)$. Correspondingly, for finite kernel power $\langle K(\mathbf{x}, \mathbf{x}) \rangle_{\mathbf{x} \sim p(\mathbf{x})}$, the eigenvalues must scale with dimension like $\eta_K \sim \mathcal{O}_D(D^{-K})$[34,53]. Examples include the widely used Gaussian kernel and dot product kernels such as NTK, which we discuss below and in Supplementary Note 4.

This scaling from the degeneracy allows us to consider multiple $P, D \to \infty$ limits leading to different learning stages. We consider a separate limit for each degenerate eigenvalue $L$ while keeping $\alpha \equiv P/N(D, L)$ finite. With this setting, we evaluate Eq. (4) with

definitions $\bar{\eta}_K \equiv N(D,K)\eta_K$, $\bar{w}_K^2 \equiv \frac{1}{N(D,K)}\sum_\rho \bar{w}_{K,\rho}^2$, to obtain the generalization error in learning stage $L$:

$$E_g^{(L)}(\alpha) = \bar{\eta}_L \bar{w}_L^2 \left( \frac{\tilde{\kappa}^2}{(\tilde{\kappa}+\alpha)^2 - \alpha} + \frac{\tilde{\sigma}_L^2 \alpha}{(\tilde{\kappa}+\alpha)^2 - \alpha} \right) + \sum_{K>L} \bar{\eta}_K \bar{w}_K^2,$$

$$\tilde{\kappa}(\alpha) = \frac{1}{2}(1 + \tilde{\lambda}_L - \alpha) + \frac{1}{2}\sqrt{(\alpha + 1 + \tilde{\lambda}_L)^2 - 4\alpha}, \qquad (9)$$

$$\tilde{\sigma}_L^2 \equiv \frac{\sigma^2 + E_g^{(L)}(\infty)}{\bar{\eta}_L \bar{w}_L^2}, \qquad \tilde{\lambda}_L \equiv \frac{\lambda + \sum_{K>L} \bar{\eta}_K}{\bar{\eta}_L}.$$

Several observations can be made:

(1) We note that $E_g^{(L)}(0) = \bar{\eta}_L \bar{w}_L^2 + \sum_{K>L} \bar{\eta}_K \bar{w}_K^2 = \bar{\eta}_L \bar{w}_L^2 + E_g^{(L)}(\infty)$. In the learning stage $L$, generalization error due to all target modes with $K < L$ has already decayed to zero. As $\alpha \to \infty$, $K = L$ modes of the target function are learned, leaving $K > L$ modes. This illustrates an inductive bias towards learning target function modes corresponding to higher kernel eigenvalues.

(2) $E_g^{(L)}(\alpha) - E_g^{(L)}(\infty)$ reduces, up to a constant $\bar{\eta}_L \bar{w}_L^2$, to the generalization error in the band-limited case, Eq. (7), with the identification of an effective noise parameter, $\tilde{\sigma}_L$, and an effective ridge parameter, $\tilde{\lambda}_L$. Inspection of $\tilde{\sigma}_L$ reveals that target modes with $K > L$ ($E_g^{(L)}(\infty)$) act as noise in the current stage. Inspection of $\tilde{\lambda}_L$ reveals that kernel eigenvalues with $K > L$ act as a regularizer in the current stage. The role of the number of eigenvalues in the white band-limited case, $N$, is played here by the degeneracy $N(D, L)$.

(3) Asymptotically, first term in $E_g^{(L)}(\alpha)$ is monotonically decreasing with $\alpha^{-2}$, while the second term shows non-monotonic behavior having a maximum at $\alpha = 1 + \tilde{\lambda}_L$. Similar to the white band-limited case, generalization error diverges due to variance explosion at $\alpha = 1 + \tilde{\lambda}_L$ when $\tilde{\lambda}_L = 0$ (a band-limited spectrum is possible) implying again a first order phase transition. Non-monotonicity caused by the noise term implies a possible peak in the generalization error in each learning stage. A phase diagram can be drawn, where phase boundaries are again defined by Eq. (8) evaluated with the effective ridge and noise parameters, Fig. 5a.

(4) Similar to the white band-limited case, optimal regularization happens when

$$\tilde{\lambda}_L = \tilde{\sigma}_L^2, \qquad (10)$$

minimizing $E_g^{(L)}(\alpha)$ for a given $\tilde{\sigma}_L$ for all $\alpha$. This result extends the previous findings on linear regression[46] to the large class of rotation invariant kernels.

(5) When all stages are considered, it is possible to observe learning curves with multiple descents with at most one peak per stage. The presence and size of the descent peak depends on the level of noise in the data and the effective regularization as shown in Eq. (8) and Eq. (9). Similar observations of multiple peaks in the learning curves were made before[36] in the context of ridgeless regression on polynomial kernels.

As an example of the effect of kernel spectrum on double-descent, consider a power law $\bar{\eta}_K = K^{-s}$ where $s \geq 1$. Then $\tilde{\lambda}_L = L^s(\zeta(s, L) + \lambda) - 1 \approx \frac{L}{s-1} + \lambda L^s$, $(L \gg 1)$, where $\zeta(s, L)$ is Hurwitz-Zeta function. In the ridgeless $\lambda = 0$ case, faster decaying spectra (higher $s$, smaller $\tilde{\lambda}_L$) are more prone to double-descent than the slower ones (Fig. 5a). Furthermore, we also observe that higher modes (higher $L$, higher $\tilde{\lambda}_L$) are more immune to overfitting, signaled by non-monotonicity, than the lower modes.

We apply our theory to Gaussian RBF regression on synthetic data in Fig. 5 where Fig. 5b demonstrates a perfect agreement with theory and experiment on Gaussian RBF with synthetic data when no label noise is present. The vertical dashed lines represent the locations where $P = N(D, 1)$ and $P = N(D, 2)$, respectively. Figure 5c shows the regression experiment with the parameters $(\tilde{\sigma}_1^2, \tilde{\lambda}_1)$ indicated by colored squares on the phase diagram (Fig. 5a). When the parameters chosen on the yellow dashed line in Fig. 5a, corresponding to the optimal regularization for fixed effective noise, the lowest generalization error is achieved in the first learning episode without a double-descent. Finally, Fig. 5d shows the theory and experiment curves with the parameters $(\tilde{\sigma}_1^2, \tilde{\lambda}_1)$ shown by the colored circles in Fig. 5a. As expected, for fixed effective regularization, increasing noise hurts generalization. For further experiments see Supplementary Note 4.

**Dot product kernels, NTK and wide neural networks.** Our theory allows the study of generalization error for trained and wide feedforward neural networks by exploiting a correspondence with kernel regression. When weights in each layer are initialized from a Gaussian distribution $\mathcal{N}(0, \sigma_W^2)$ and the size of hidden layers tend to infinity, the function $f(\mathbf{x}, \boldsymbol{\theta})$ learned by training the network parameters $\boldsymbol{\theta}$ with gradient descent on a squared loss to zero training error is equivalent to the function obtained from ridgeless ($\lambda = 0$) kernel regression with the NTK: $\mathbf{K}_{\mathrm{NTK}}(\mathbf{x}_i, \mathbf{x}_j) = \nabla_{\boldsymbol{\theta}} f(\mathbf{x}_i, \boldsymbol{\theta}_0) \cdot \nabla_{\boldsymbol{\theta}} f(\mathbf{x}_j, \boldsymbol{\theta}_0)$[13]. For fully connected neural networks, the NTK is a dot product kernel $K_{\mathrm{NTK}}(\mathbf{x}, \mathbf{x}') = K_{\mathrm{NTK}}(\mathbf{x} \cdot \mathbf{x}')$[13,34]. For such kernels and spherically symmetric data distributions $p(\mathbf{x}) = p(||\mathbf{x}||)$, kernel eigenfunctions do not have a radial part, and consequently the eigenvalues are free of a $z$-index. Therefore, $k$-th eigenvalue has degeneracy of the degree $k$ spherical harmonics, $\mathcal{O}_D(D^k)$, ($K, L \to k$, $l$ and $\rho \to m$)[34], allowing recourse to the same scaling we used to analyze rotation invariant kernels in the previous section. The learning curves for infinitely wide neural network will thus have the same form in Eq. (9), evaluated with NTK eigenvalues and with $\lambda = 0$.

In Fig. 6a, we compare the prediction of our theoretical expression for $E_g$, Eq. (4), to NTK regression and neural network training. The match to NTK training is excellent. We can describe neural network training up to a certain $P$ after which the correspondence to NTK regression breaks down due to the network's finite-width. For large $P$, the neural network operates in under-parameterized regime where the network initialization variance due to finite number of parameters starts contributing to the generalization error[3,38,42,54]. A detailed discussion of these topics is provided in Supplementary Note 4.

Neural networks are thought to generalize well because of implicit regularization[2,51,52]. This can be addressed with our formalism. For spherical data, we see that the implicit regularization of a neural network for each mode $l$ is given by $\tilde{\lambda}_l = \frac{\sum_{k>l} \bar{\eta}_k}{\bar{\eta}_l}$. As an example, we calculate the spectrum of NTK for rectifier activations, and observe that the spectrum whitens with increasing depth[52], corresponding to larger $\tilde{\lambda}_l$ and therefore more regularization for small learning stages $l$ (Fig. 6b). The trend for small degree harmonics $l$ is especially relevant since, as we have shown, approximately $D^l$ samples are required to learn degree $l$ harmonics. In this small $l$ regime, we see that deep networks exhibit much higher effective regularization compared to shallow ones due to slower decay of eigenvalues.

**Discussion**

We studied generalization in kernel regression using statistical mechanics and the replica method[33]. We derived an analytical expression for the generalization error, Eq. (4), valid for any kernel and any dataset. We showed that our expression explains generalization on real datasets, and provided a detailed analysis of

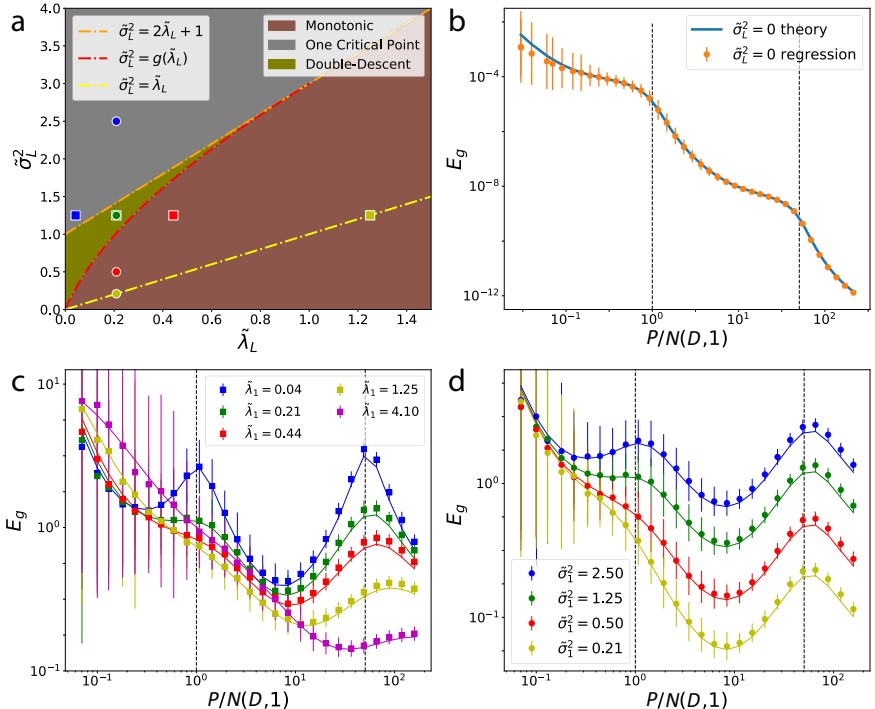

**Fig. 5 Gaussian RBF kernel regression on high-dimensional spherical data. a** Phase diagram for non-monotonic learning curves obtained from the theory by counting the zeros of $\frac{\partial E_g}{\partial \alpha}$. Colored squares and colored circles correspond to curves in **c**, **d**, respectively. **b** Kernel regression with Gaussian RBF $K(\mathbf{x}, \mathbf{x}') = e^{-\frac{1}{2D\omega^2}||\mathbf{x}-\mathbf{x}'||^2}$ with $\omega = 3$, $D = 100$ and noise-free labels. Target is $\bar{f}(\mathbf{x}) = \sum_{k,m} \bar{w}_{km} \sqrt{\eta_{km}} Y_{km}(\mathbf{x})$ with random and centered weights $\bar{w}_{km}$ such that $\langle \bar{w}_{km}^2 \rangle = \eta_{km}$ (Supplementary Note 5). Dashed lines represent the locations of $N(D, 1)$ and $N(D, 2)$, showing different learning stages. **c, d** Generalization error for Gaussian RBF kernel for various kernel widths $\omega$ corresponding to specific $\tilde{\lambda}_L$'s and noise variances $\tilde{\sigma}_L$ pointed in the phase diagram in $D = 100$. Solid lines—theory (Eq. (4)). Larger regularization suppresses the descent peaks, which occur at $P^* \sim N(D, L)$ shown by the vertical dashed lines. **c** Varying $\tilde{\lambda}_L$ with fixed the $\tilde{\sigma}_L$. **d** vice versa. For fixed noise, we observe an optimal $\tilde{\lambda}_1$ for up to $P/N(D, 1) \sim 10$ after which the next learning stage starts. Error bars indicate standard deviation over 300 trials for **b** and 100 trials for **c, d**.

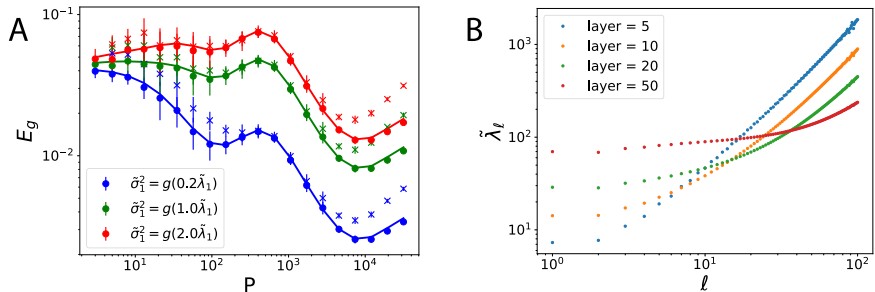

**Fig. 6 Comparison of our theory with finite width neural network experiments. a** 2-layer NTK regression and corresponding neural network training using NeuralTangents package[55] with 50000 hidden units for $D = 25$ with varying noise levels chosen according to $g(\lambda)$. Target function is a single degree mode $\bar{f}(\mathbf{x}) = c_k Q_k^{(D-1)}(\boldsymbol{\beta} \cdot \mathbf{x})$, where $c_k$ is a constant, $\boldsymbol{\beta}$ is a random vector, and $Q_k^{(D-1)}$ is the $k$-th Gegenbauer polynomial (see Supplementary Note 5 and 6). Here we picked $k = 1$ (linear target). Solid lines are the theory predicted learning curves (Eq. (4)), dots represent NTK regression and × represents $E_g$ after neural network training. Correspondence between NN training and NTK regression breaks down at large sample sizes $P$ since the network operates in under-parameterized regime and finite-size effects become dominating in $E_g$. Error bars represent standard deviation of 15 averages for kernel regression and 5 averages for neural network experiments. **b** $\tilde{\lambda}_l$ dependence to mode $l$ across various layer NTKs. The weight and bias variances for the neural network are chosen to be $\sigma_W^2 = 1$ and $\sigma_b^2 = 0$, respectively.

its application to band-limited kernels with white spectra and the widely used class of rotation invariant kernels[39,40] operating on spherical data. For the latter case, we defined an effective regularization and an effective noise which govern the generalization behavior, including non-monotonicity of learning curves. It will be interesting to see if analogues of these concepts can be defined for real datasets. Our results are directly applicable to infinite-width limit of neural networks that admit a kernel description (including feedforward, convolutional and recurrent neural

networks)[13,55–59], and explain their inductive bias towards simple functions[35,51,60–64]. We also note a closely related recent study[19], which we discuss further in Supplementary Discussion, that utilizes random matrix theory to study generalization in kernel regression.

One goal of our present work is to provide a framework that incorporates structural information about the data distribution into a realistic prediction of generalization performance that holds for real data and any kernel. Indeed, a recent study

suggested that structure in data allows kernel methods to out-perform pessimistic generalization expectations based on the high ambient dimension[65]. Authors of a different study[66] calculated test error of random Fourier features model using random matrix theory techniques without strong assumptions on data distribution and obtained excellent agreement on real datasets. Overall, our results demonstrate how data and inductive biases of a model interact to shape generalization behavior, and in particular the importance of the compatibility of a learning task with the model for sample-efficient learning. Our findings elucidate three heuristic principles for generalization in kernel regression.

First is the spectral bias. The eigendecomposition of the kernel provides a natural ordering of functions which are easiest to estimate. Decomposing generalization error into modal errors, we found that errors in spectral modes with large eigenvalues decrease more rapidly with increasing sample size than modes with small eigenvalues, also observed in[34], illustrating a preference to fit certain functions over others. Our findings are consistent with other experimental results and analytical ones which derive error bounds on test risk to elucidate the spectral or frequency bias of NTK and NNGPK[67-70].

Consequently, how a given task decomposes in the eigenbasis, a heuristic that we name task-model alignment, determines the number of samples required to achieve good performance: tasks with most of their power in top eigenmodes can be learned in a sample-efficient manner. We introduced cumulative power distribution as a metric for task-model alignment and proved that target functions with higher cumulative power distributions will have lower normalized generalization error for all $P$ under the same kernel and data distribution. A related notion of kernel compatibility with target was defined in[71,72], which we discuss in detail in Supplementary Discussion.

The third phenomenon we explore is how non-monotonicity can appear in the learning curves when either labels are noisy, or the target function has modes that are not expressible with the kernel. Non-monotonicity is caused by the variance term in the bias-variance decomposition of the generalization error. In the analytically tractable models we considered, this is related to a phase transition appearing in separate learning stages for the rotation invariant kernels. Non-monotonicity can be mitigated with explicit or implicit regularization[38,42,73]. We showed the existence of an optimal regularizer, independent of sample size, for our theoretical settings. When applied to linear regression, our optimal regularizer matches that previously given in literature[46].

Non-monotonicity in generalization error gathered a lot of interest recently. Many studies pointed to absence of overfitting in overparameterized machine learning models, signaled by a peak and a subsequent descent in generalization error as the model complexity, or the number of parameters, increases, and the model transitions from an under-parameterized to over-parameterized (interpolating) regime[3,5,35,38,42,43,54,66,74,75]. Multiple peaks are also possible in this context[76]. Our work provides an explanation for the lack of overfitting in overparameterized models by elucidating strong inductive biases of kernel regression, valid even in the interpolation limit, which includes infinitely overparameterized limits of neural networks. Sample-wise non-monotonicity has also been observed previously in many models[5,24,25,54,73], including ones that show multiple peaks[36,43,46,77]. A closely related study obtained an upper bound for test risk in ridgeless regression which shows non-monotonic behavior with increasing sample size whenever $P \sim \mathcal{O}(D^L)$, consistent with our results on rotation invariant kernels and isotropic data.

An interesting comparison can be made between the multiple peaks we observed in our analytically solvable models and the multiple peaks observed in random features models[43,76]. In these

models, one of the peaks (termed "nonlinear" in a previous study[43]) happens when the number of samples reaches the number of features, and thus the number of parameters of the model, crossing the interpolation threshold. While the peak we observed in the white band-limited case with nonlinear features also happens at the interpolation threshold ($P = N$), the mechanisms causing double descent are different. In random features models, this peak is due to variance in the initialization of the random feature vectors. In our example, such variance does not exist. The peak is due to overfitting the noisy labels and disappears when there is no noise. The peaks observed for the rotationally invariant case has a more subtle connection. In each learning stage, peaks occur when number of samples reach the degeneracy of eigenfunctions in that stage. While kernel regression is non-parametric, one can think of this again as crossing an interpolation threshold defined by the dimensionality of the feature space in the large-$D$ limit. Like the white band-limited case, these peaks are due to overfitting noise.

While our theory is remarkably successful in its predictions, it has limitations. First, the theory requires eigendecomposition of the kernel on the full dataset which is computationally costly. Second, its applicability to state-of-the-art neural networks is limited by the kernel regime of networks, which does not capture many interesting and useful deep learning phenomena[62,78]. Third, our theory uses a Gaussian approximation[79] and a replica symmetric ansätz[33]. While these assumptions were validated by the remarkable fit to experiments, it will be interesting to see if their relaxations reveal new insights.

## Methods

**Statistical mechanics formulation.** With the setting described in the main text, kernel regression problem reduces to minimization of the energy function

$$H(\mathbf{w}) \equiv \frac{1}{2\lambda} \sum_{\mu=1}^{P} \left( \sum_{\rho=1}^{N} (\bar{w}_\rho - w_\rho) \psi_\rho(\mathbf{x}^\mu) + \epsilon^\mu \right)^2 + \frac{1}{2} ||\mathbf{w}||_2^2. \quad (11)$$

The quantity of interest is the generalization error in Eq. (2), which in matrix notation is

$$E_g = \left\langle (\mathbf{w}^* - \bar{\mathbf{w}})^\top \Lambda (\mathbf{w}^* - \bar{\mathbf{w}}) \right\rangle_{\mathcal{D}}, \quad (12)$$

where $\Lambda_{\rho\gamma} = \eta_\rho \delta_{\rho\gamma}$ represents a diagonal matrix with entries given by the RKHS eigenvalues $\eta_\rho$.

In order to calculate the generalization error, we introduce a Gibbs distribution $p_G(\mathbf{w}) \equiv \frac{1}{Z} e^{-\beta H(\mathbf{w})}$ with the energy function in Eq. (11). In the $\beta \to \infty$ limit, this Gibbs measure is dominated by the solution to the kernel regression problem. We utilize this fact to calculate the generalization error for kernel regression. This can be done by introducing a source term with strength $J$ to the partition function,

$$Z(J) = \int d\mathbf{w} e^{-\beta H(\mathbf{w}, \mathcal{D}) + J\beta P(\mathbf{w} - \bar{\mathbf{w}})^\top \Lambda (\mathbf{w} - \bar{\mathbf{w}})}, \quad E_g(\mathcal{D}) = \lim_{\beta \to \infty} \frac{1}{\beta P} \frac{d}{dJ} \ln Z(J) \Big|_{J=0}, \quad (13)$$

where we recognize the free energy $\beta F \equiv -\ln Z(J)$ which is the relevant quantity to compute generalization error for a given dataset, $E_g(\mathcal{D})$. In Supplementary Note 2, we introduce other source terms to calculate training error, average estimator and its variance.

The free energy depends on the sampled dataset $\mathcal{D}$, which can be thought of as a quenched disorder of the system. Experience from the study of physics of disordered systems suggests that the free energy concentrates around its mean (is self-averaging) for large $P$[33]. Therefore, we calculate the typical behavior of the system by performing the average free energy over all possible datasets: $\beta F = \beta \langle F \rangle_{\mathcal{D}} = -\langle \ln Z(J) \rangle_{\mathcal{D}}$ in the $P \to \infty$ limit.

All calculations are detailed in Supplementary Notes 1 and 2. Here we provide a summary. To perform averages over the quenched disorder, we resort to the replica trick[80] using $\langle \log Z(J) \rangle_{\mathcal{D}} = \lim_{n \to 0} \frac{1}{n} (\langle Z(J)^n \rangle_{\mathcal{D}} - 1)$. A key step is a Gaussian approximation to the average over the dataset in the feature space[81], which exploits the orthogonality of the feature vectors with respect to the input distribution $p(\mathbf{x})$. These averages are expressed in terms of order parameters defining the mean and the covariance of the Gaussian. The calculation proceeds by a replica symmetric ansätz[33], evaluating the saddle point equations and taking the $\beta \to \infty$ limit.

**Modal errors**. The generalization error can be written as a sum of modal errors arising from the estimation of the coefficient for eigenfunction $\psi_\rho$:

$$E_g = \sum_\rho \eta_\rho \overline{w}_\rho^2 E_\rho, \tag{14}$$

where

$$E_\rho = \frac{1}{\overline{w}_\rho^2} \left\langle (w_\rho^* - \overline{w}_\rho)^2 \right\rangle_{\mathcal{D}} = \frac{1}{1-\gamma} \frac{\kappa^2}{(\kappa + P\eta_\rho)^2}. \tag{15}$$

We now provide a proof that the mode error equation implies that the logarithmic derivatives of mode errors have the same ordering as the kernel eigenvalues when $\sigma = 0$. Assuming that $\eta_\rho > \eta_{\rho'}$, and noting that $\kappa'(P) = -\frac{\kappa\gamma}{P(1+\gamma)} < 0$ since $\kappa, \gamma, P > 0$, we have

$$\frac{d}{dP} \log\left(\frac{E_\rho}{E_{\rho'}}\right) = -2\left[\frac{\kappa'(P) + \eta_\rho}{\kappa + P\eta_\rho} - \frac{\kappa'(P) + \eta_{\rho'}}{\kappa + P\eta_{\rho'}}\right] < 0. \tag{16}$$

Thus, we arrive at the conclusion

$$\frac{d}{dP}\log E_\rho < \frac{d}{dP}\log E_{\rho'} \quad\Rightarrow\quad \frac{1}{E_\rho}\frac{dE_\rho}{dP} < \frac{1}{E_{\rho'}}\frac{dE_{\rho'}}{dP}. \tag{17}$$

Let $u_{\rho,\rho'}(P) = \log\left(\frac{E_\rho}{E_{\rho'}}\right)$. The above derivation demonstrates that $\frac{d}{dP}u_{\rho,\rho'}(P) < 0$ for all $P$. Since $u_{\rho,\rho'}(0) = 0$, this implies that $u_{\rho,\rho'}(P) < 0$ or equivalently $E_\rho < E_{\rho'}$ for all $P$. This result indicates that the mode errors have the opposite ordering of eigenvalues, summarizing the phenomenon of spectral bias for kernel regression: the generalization error falls faster for the eigenmodes with larger eigenvalues. If the target function has norm $T = \left\langle \overline{f}(\mathbf{x})^2 \right\rangle = \sum_\rho \eta_\rho \overline{w}_\rho^2$ then the generalization error is a convex combination of $\{TE_\rho(P)\}_{\rho=1}^\infty$. The quantities $TE_\rho(P)$ maintain the same ordering as the normalized mode errors $E_\rho$ for all $P$, and we see that re-allocations of target function power that strictly increase the cumulative power distribution $C(\rho) = \frac{1}{T}\sum_{\rho' \le \rho} \eta_{\rho'} \overline{w}_{\rho'}^2$ curve must cause a reduction in generalization error. We emphasize that, for a fixed set of kernel eigenvalues, strictly higher $C(\rho)$ yields better generalization but provide a caveat: for a fixed target function, comparison of different kernels requires knowledge of both the change in the spectrum $\eta_\rho$ as well as changes in the $C(\rho)$ curve.

**Diagonalizing the kernel on real datasets**. Calculation of $E_g$ requires two inputs: kernel eigenvalues $\eta_\rho$ and teacher weights $\overline{w}_\rho$, both calculated using the underlying data distribution. For a dataset with a finitely many samples, we assume a discrete uniform distribution over data $p(\mathbf{x}) = \frac{1}{M}\sum_{i=1}^M \delta(\mathbf{x} - \mathbf{x}_i)$ with $M$ being the size of the whole dataset (train+test). Then, the corresponding eigenvalue problem reads:

$$\eta_k \phi_k(\mathbf{x}') = \int p(\mathbf{x}) K(\mathbf{x}, \mathbf{x}') \phi_k(\mathbf{x}) d\mathbf{x} = \frac{1}{M}\sum_{i=1}^M K(\mathbf{x}_i, \mathbf{x}') \phi_k(\mathbf{x}_i). \tag{18}$$

Given a kernel $K(\mathbf{x}, \mathbf{x}')$, one can evaluate the $M \times M$ kernel Gram matrix $\mathbf{K}_{ij} = K(\mathbf{x}_i, \mathbf{x}_j)$ and solve for the eigenvalues $\mathbf{\Lambda}_{kl} = \eta_k \delta_{kl}$ and eigenfunctions $\mathbf{\Phi}_{ki} = \phi_k(\mathbf{x}_i)$ of $\mathbf{K} = N\mathbf{\Phi}\mathbf{\Lambda}\mathbf{\Phi}^\top$. Note that both data indices and eigen-indices take values $i, k = 1,...,M$. For a target function with multiple classes $\overline{f}(\mathbf{x}) : \mathbb{R}^D \to \mathbb{R}^C$, we denote the one-hot encoded labels $\mathbf{Y} = [\mathbf{y}_1, ..., \mathbf{y}_C] \in \mathbb{R}^{M \times C}$ and obtain the teacher weights for each class with $\overline{\mathbf{w}}_c = \frac{1}{M}\mathbf{\Lambda}^{-1/2}\mathbf{\Phi}^\top \mathbf{y}_c$. For solving kernel regression, each of the $C$ one-hot output channels can be treated as an individual target function $f_{t,c}(\mathbf{x})$ where $f_{t,c}(\mathbf{x}^\mu) = y_c^\mu$ for one-hot labels $y_c^\mu$. The weights $\overline{\mathbf{w}}_c$ obtained above allows the expansion of each teacher channel in the kernel's eigenbasis $f_{t,c}(\mathbf{x}) = \sum_{k=1}^M \overline{w}_{c,k}\sqrt{\eta_k}\phi_k(\mathbf{x})$. The total generalization error for the entire task is simply $E_g = \sum_{c=1}^C \left\langle (f_c^*(\mathbf{x}) - f_{t,c}(\mathbf{x}))^2 \right\rangle_{\mathbf{x},\mathcal{D}}$ where $f_c^*$ is the kernel regression solution for output channel $c$. Note that, with the choice of one-hot labels, the total target power is normalized to 1. After computing learning curves for each channel $c$, which requires plugging in $\{(\eta_k, \overline{w}_{c,k}^2)\}_{k=1}^M$ into our theory, the learning curves for each channel are simply summed to arrive at the final generalization error.

In other cases, when we do not generally possess a priori knowledge about $p(\mathbf{x})$, the underlying data distribution, solving the kernel eigenvalue problem in Eq. (3) appears intractable. However, when we are provided with a large number of samples from the dataset, we may approximate the kernel eigenvalue problem by using a Monte-Carlo estimate of the data density i.e. $p(\mathbf{x}) \approx \frac{1}{M}\sum_{i=1}^M \delta(\mathbf{x} - \mathbf{x}_i)$ with $M$ being the size of the dataset. Then Eq. (18) approximates the eigenvalues and eigenvectors where one obtains the exact eigenvalues when the number of samples is large, $M \to \infty$[40].

## Data availability
All data generated and/or analyzed in this study, along with the analysis code, are available in the Github repository, https://github.com/Pehlevan-Group/kernel-generalization.

## Code availability
All code used to perform numerical experiments, analyze data and generate figures are available in the Github repository, https://github.com/Pehlevan-Group/kernel-generalization.

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

## Acknowledgements

We thank the Harvard Data Science Initiative and Harvard Dean's Competitive Fund for Promising Scholarship for their support. We also thank Jan Drugowitsch and Guillermo Valle-Perez for comments on the manuscript.

## Author contributions

A.C., B.B., and C.P. contributed to the planning of the study. A.C. and B.B. performed the analytical calculations and numerical simulations. A.C., B.B., and C.P. contributed to the interpretation of the results and writing of the manuscript.

## Competing interests

The authors declare no competing interests.

**Additional information**

