## [Peer Review File · Nature Communications]

Reviewer #1 (Remarks to the Author):

Main contributions :

This paper presents an analytical derivation of the generalization performance of kernel regression in a very general setup (applicable to any kernel and any data distribution). The authors clearly show how one can predict performance by studying the interaction between the inductive bias of the kernel (i.e. its eigendecomposition in a basis of orthogonal functions) and the task at play (i.e. the decomposition of the data distribution in this basis).

The authors first show results on classic vision datasets (MNIST and CIFAR10), giving intuition on the kernel-task alignment, then turn to two more theoretical setups. First, the band-limited kernel RKHS setup, which includes linear regression, in which case they obtain a simple formula for the test error enabling a detailed study of the presence of a non-monotonicity in the test error in the (λ, σ) phase-space. Second, the rotationally-invariant setup, where the degeneracy of the eigenmodes enables to reuse results from the band-limited setup in multiple "learning episodes", corresponding the different degenerate eigenmodes.

The mathematical derivation of the generalization error is sound and elegant, albeit not fully rigorous in two ways : (i) it relies on a gaussian approximation, and (ii) it uses the Replica Method from statistical physics. However, I believe the strength of the expression obtained, and the excellent agreement with numerics fully justify the use of such methods.

Overall, the text is very well written, the figures are extremely clear, and the results are significantly more general than previous efforts. The authors put significant emphasis on making the results understandable: the phase spaces of figs 3 and 4 are excellent, and the effective noise and regularization of eq. 9 give a very intuitive understanding of the multiple descent curve (which was rather obscurely presented in previous work). For all these reasons, I think this paper is of strong interest to the community.

Literature review :

In my opinion, the literature review needs to be substantially extended. The paper needs a clearer discussion of other works discussing multiple non-monotonicities, which in the current state are cited without much details [1,2,3], or missed out [4,5,6]. In particular, [1], which obtains a very similar multiple descent curve (in a more limited setup), deserves more discussion. [6] is the most related paper studying kernel regression on real data (but only with random Fourier features) and crucially needs to be cited.

I would separate these references into those with several parameter-wise nonmonotonicities [5] and those with several sample-wise nonmonotonicities [1,2,3,4].

It would also be helpful to put the sentence appearing in the conclusion -- "Our results extend the finding on the former type of peak, demonstrating the possible existence of many error peaks at each $P = O(D^K)$. The non-monotonicity we observed in the white band-limited RKHS is due to a peak of the latter type" in the introduction. Here, the terminology of linear and nonlinear peak introduced in [3] could be helpful.

- [1] Liang et al. 2020, Multiple Descent of minimum-norm interpolants
- [2] Nakkiran et al. 2020, Optimal Regularization can mitigate double descent
- [3] d'Ascoli et al. 2020, Triple Descent and the two kinds of overfitting
- [4] Chen et al. 2020, Design your own generalization curve
- [5] Pennington et al. 2020, The Neural Tangent Kernel in high dimensions
- [6] Couillet et al. 2020, A random matrix analysis of random Fourier features

Comments :

I had a hard time understanding was whether the results are valid at finite size, both for D and for P . For D : I thought eq. 9 was valid only for infinite D but in fig. SI.3 the authors seem to say there is a finite and infinite version of the formula (the dashed lines of panels A and B respectively). For P : the fact that P appears for the scaling makes things confusing. If the formula is valid for

infinite P , how come it works so well even for P of order 10 in figs 2 and 5 ? It would be nice to have a clearer discussion on the domain of validity of each formula. In fact, more emphasis could be put on the fact that the domain of validity seems to range down to very small values $P=10$, which is very surprising ! How many runs are averaged to obtain such neat curves ?

More discussion could be given on the double descent observed for CIFAR10 experiment : what determines the location of the peaks and why are they shifted as noise increases ?

One slight concern is in Figure 5B. For small values of P , the experimental and theoretical points do not match very well when the variance becomes large, despite the network operating in the over-parametrized regime. Even though the error bars of the numerical curve seem to just about reach the analytical curve (not clear at very low P), they are systematically biased, signalling that there is a non-captured phenomenon at play. Is this simply a finite-size effect, or something more profound ?

Minor comments :

There are a few notational mixups. In Methods / Diagonalizing the kernel on real datasets : N is the size of the dataset instead of P .

SI : M seems to denote what is called N in main text.

Conclusion : "the number of parameters P reach...". This typo urgently needs correction as it can be very misleading !

Fig 2 : x axis should be P

Fig 3 : x axis is α

Fig SI.3 : missing the vertical lines

Line 54 : "generalisation in"

Fig SI.4 : why only $D=20$? Can't we make D bigger to have less finite-size effects ? Or plot for increasing D to see the gaps "fill in" ?

Reviewer #2 (Remarks to the Author):

Summary of the paper: This paper provides a theoretical formulation, using methods of statistical physics, of generalization error in kernel regression and infinite-width neural networks.

Outlined contribution of the work from the authors are:

[C1] Analytic expression for generalization error applicable to any kernel or data distribution (Eq 4): from using techniques from statistical physics

[C2] Application to this theory to 'real' and 'synthetic' datasets for various kernels. Which covers the infinite-width limit of deep neural networks.

[C3] Inductive bias of kernel regression towards 'simple functions' characterized by spectral properties of the kernel

[C4] Show more data could hurt generalization if noisy, leading to non-monotonic learning curves.

[C5] Study broad class of rotation invariant kernels, on spherical uniform data and large input dimensions which allow detailed mathematical analysis.

Significance:

Understanding generalization of overparameterized models in machine learning is a fundamental question which is an active research area. Traditional uniform convergence bounds often does not provide significant insight into how modern overparameterized models generalized well to unseen data. Therefore new techniques to understand "typical" generalization property instead of "worst" case is becoming increasingly more important. This paper uses techniques from statistical physics (replica method) allowing such analysis.

I think this work can be interesting to the general scientific community 1) since machine learning and AI techniques are becoming ubiquitous in the field of science 2) this work addresses important

“scientific questions” for important complex learning systems.

I believe provided theory is an important step towards understanding generalization in machine learning and in particular overparameterized neural networks. Experiments in this paper showing various generalization phenomena demonstrate one could obtain insights on inner workings of how learning is happening in kernel type models.

Given that, there are some limitations I see (also detailed more in [Technical concerns] below).

First the claim that “our theory can predict which kernels or neural architectures are well suited” seems a little too strong as what proposed theory actually could do. If my understanding regarding the target function in [technical concerns] is correct, analytic expression for generalization is limited to very specific target functions, and unclear whether it applies to natural labels that exist in the real dataset.

Moreover, the current theory requires “eigen decomposition” of the full dataset to obtain kernel eigenvalues and eigenvectors. With similar or even less compute, one could easily do either small k-fold cross validation or leave-one-out cross validation on a given dataset to determine how well a given kernel/neural architecture can perform on a given dataset.

Lastly, there are few concerns whether NTK-style kernel learning captures the good generalization property of SOTA finite networks. There are many studies regarding “feature learning” vs “lazy learning” and various finite network training techniques (large learning rate, l2 regularization) that diverts out of vanilla NTK setting. Moreover while many of the analysis is based on rotationally invariant kernels, widely successful convolutional models are not exactly rotationally invariant in the sense defined in this paper. Certainly these are beyond the scope of this paper and would be an interesting follow-up study. I believe, in general, clearly stating current limitations and what future research is required would be service to the research community.

Novelty:

- The Replica method is a common technique used widely in the theoretical physics community. Even in the context of machine learning, there were quite a few works utilizing similar techniques to obtain learning curves for kernel based methods (SVM, GP). While there are citations to few of these works, the paper does not clearly state how this submission distinguishes itself.
- One thing that was not very clearly stated was whether the used method by the author is novel or not. Authors do cite their ICML 2020 publication by the same set of authors (Ref 31) which bulk of the content and contributions is quite similar to this submission. For example, both use a replica method to obtain learning curves of kernel regression and use connection to infinitely-wide neural networks to make statements about overparameterized networks ([C1, C2]). Also similar spectral dependency and exact analysis based on data on the sphere is done ([C3]). To the reviewer, it was not clear whether this work is an “expansion” of Ref 31 for a journal submission or novel orthogonal work. I strongly encourage authors to make this point clear to set the readers' expectations straight.
- In regards to the previous work, I was hoping to find out differences in method/result of [41] (Cohen et al., 2019) since they also use replica methods to obtain learning curves for GP and make connections to infinitely-wide networks. In general, it is hard to tell which contribution in this paper is novel / different from previous work. If spaces are allowed, or at least in the supplement information, I encourage the authors to expand related work and how the result of this paper distinguishes among other works.

Technical concerns:

- One potential drawback of the theory is that strong assumptions on the target function. Theory always assumes that the target function lies in RKHS and is able to expand in terms of the

eigenbasis.

Under some smooth assumption on the labels of the dataset this may be true. However, for a typical image classification task with one-hot labeling, the target function is extremely not smooth and I expect it is either hard to apply theory in this paper or agreement between theory and experiment wouldn't be as good as shown. If I'm not mistaken, the paper should explicitly clarify the limitation, unless it can be misleading.

This is especially true, since the authors boast application to the "real" dataset. Please correct me if I'm mistaken, however, from my reading the test data label always needs to come from "target function \bar{w} " (inferred from training labels) even in the MNIST or CIFAR10 experiments. I believe the actual test set label is quite different from the label assigned by this target function from \bar{w} .

In other words, the theory assumes the target is within the class the models can learn. Often real tasks would have quite complicated task functions, while functions represented by simple kernel or neural architecture are quite far. This will lead to not as good generalization but current setup, to my understanding, does not capture the case.

While I believe understanding the learning curve in this setting is an important problem and can lead to understanding of very important questions of how learning/generalization occurs. There still seems to be quite a gap in understanding real target function.

- The authors obtain a solution assuming replica symmetry. Most cases, this is a fine assumption and often a good interesting starting point of analysis. This can be also retrospectively validated by empirical check that assumption was good. So I believe this is not a significant issue but I do wonder if the authors have considered the possibility of replica symmetry breaking (RSB) solutions. In statistical physics, this often leads to more interesting phenomenology and wonder in the context of kernel methods or infinitely wide neural networks, whether one wants/needs to consider RSB solutions to capture a more realistic setting.

Reproducibility:

I think the paper provided sufficiently enough theoretical details and derivation for anyone who has a background in kernel method and knowledgeable in statistical physics could follow the derivation. Also the paper provides enough pointers for motivated readers to learn about the background.

However on the empirical front, the provided details does not seem to be sufficient that at least the reviewer, myself, is not confident that enough details are provided to reproduce the result easily. For example, experiments in Fig 5, what was the task that's learning on, learning rate, l_2 regularization, parameterization, initialization weight/bias variances? It would benefit the readers to either 1) provide details of experiments in detail and/or 2) share code for reproducing experiments to understand proposed methods/analysis more carefully.

Nits:

L7 end: "new a theory" -> "a new theory"

Fig2 caption: noise variance $\sigma = 0, 1.5$ (5 is missing)

L47: consider replacing [10] and instead adding Matthews et al., "Gaussian Process Behaviour in Wide Deep Neural Networks", ICLR 2018. While [10] is an important reference and requires citation but probably not here.

Fig 4: (A-D) is missing in the actual figures

Section 3.2.1: For spherical dataset on NTK consider also citing:

Yang & Salman, "A Fine-Grained Spectral Perspective on Neural Networks", 2019 and Bietti & Mairal, "On the Inductive Bias of Neural Tangent Kernels", NeurIPS 2019.

Recommendation:

I believe this submission is a very solid work tackling an important problem in machine learning applying tools from theoretical physics. In general technique used is concrete and the results are convincing and illuminates various phenomenologies. There are some concerns on significance and novelty that should be clearly addressed by the authors.

Reviewer #3 (Remarks to the Author):

Review of Spectral Bias and Task-Model Alignment Explain Generalization in Kernel Regression and Infinitely Wide Neural Networks

December 9, 2020

1 Contribution

The authors of the paper present new insights on the generalization error of kernel methods. Using tools from physics, they investigate this generalization, obtaining a general formula which matches empirically the generalization error of kernel regression, even for finite number of data points. More precisely, using the fact that the minimization problem can be approximated by a statistical model and letting the temperature going to 0, the problem becomes a computation of a difficult integral. This computation is done using a Gaussian approximation, the so-called replica method and the method of steepest descent. This leads the author to a simple (yet interesting) expression for the generalization error of kernel methods. In this general framework, they find that components of the target function along the eigenfunctions of the Kernel with large eigenvalues are "fitted faster" as the number of parameters increases: this is the spectral bias property. Using this insight, they are able to quantify the difficulty of a task given a kernel. All these assertions are backed by real world experiments using the MNIST and CIFAR dataset.

Then, they apply their main formula to study the asymptotic behavior of the generalization error E_g to two settings:

1. Band Limited Kernels: these are toy models in order to study the more general setting of Rotation Invariant Kernel. When the number of data points $P \rightarrow \infty$ and the number of constant eigenvalues $N \rightarrow \infty$ with a constant ratio $\alpha = P/N$, the limiting E_g can be computed explicitly. They can thus study its behavior as a function of α , of the ridge λ and the amount of noise σ^2 . It has to be noticed that if they recover the so-called double descent, they obtain a stronger result since they manage to compute explicitly the $\lambda - \sigma^2$ phase diagram for the double descent, and the different asymptotic decays of E_g in α .

2. Rotation Invariant Kernels: one example of such Rotation Invariant Kernel, which is studied empirically and theoretically in this paper, is the 1 recent NTK for infinitely large Neural Networks. In the spectrum of Rotation Invariant Kernels, eigenvalues have multiplicities which grow as the dimension of the data D goes to infinity at different speeds. This lead the authors to define different "learning episodes" which are obtained using different asymptotics for P and D . They note that indeed such limit can be described in term of learning episodes in the sense that larger eigenvalues

are already learned and do not appear in the generalization error, and only the components along the eigenfunctions selected by the asymptotics of P and D are learned as the number of parameters increase. They also give interesting insight on the role of the smaller eigenvalues of the kernel and of the residue of the target function on the corresponding eigenfunctions during this learning episode: the eigenvalues act as regularizer as they induce an effective regularization whereas the residue of the function act as a noise term. This leads to a double descent curve during this learning episode, which leads ultimately to a multiple descent curve when considering all stages.

2 Impression

The study of generalization of kernel methods using the statistical tool box is very interesting and the key results are well illustrated by real world experiments. What is interesting in this paper is that it provides a general formula for general kernel but also, most importantly, a lot of insights and new results about the specific but important kernels which are the Rotation Invariant Kernel. It was very appreciated to have the simpler section about Band Limited Kernels since most of the intuitions are already convey in this section. The concepts, results and computational techniques are well introduced and the computations are, in my opinion, sufficiently detailed. Overall, I think the article is very interesting and very well written, reaches the level of publications in Nature Communications and therefore worthy of publication after appropriate corrections and modifications by the authors.

3 Comments

What follows are minor comments, only on the main text.

1. Correct the λ line 134 in Equation (5): it should be η .
2. In the paper “Kernel Alignment Risk Estimator: Risk Prediction from Training Data” [NeurIPS 2020] which does not study as deeply the asymptotic of the generalization error as in this paper, a similar equation as Equation (4) is proven mathematically. The κ plays the role of the Signal Capture Threshold ϑ and the $1 - \gamma$ should be linked to its derivative wrt to the ridge $\partial \lambda \vartheta$. It would be interesting to discuss the similarities and difference between these two formulas for the E_g and more generally between these two articles. 2
3. Also in your experiments, in order to use the formula (4), one has to approximate the eigenvalues of the continuous kernel using the training data, besides one has to also estimates the weights of the target label function. One can avoid this by considering the “Kernel Alignment Risk Estimator” (see previous point).
4. Since you are talking about Task-Model Alignment, could you elaborate on the link between your definition and results and the Kernel Alignment as defined for example in <http://papers.neurips.cc/paper/1946-on-kernel-target-alignment.pdf> ?
5. Correct Figure 3, the labels for the x should be α and not P . Also it not clear at all what is the horizontal line in the phase diagram (I understood it was linked with the plot C but it took me some time), it would be better actually to remove this line and to keep only the dots and to explain what are these dots.
6. Using K both for the kernel and as an index for the eigenvalues might not be a good idea.
7. In Figure 4, labels A, B, C, D are missing.

8. Same figure, the three other figures, is it $E(1)g$ or Eg which is drawn ? (since there are multiple descent, this would be in contradiction with the main text if it was $E(1)g$). Besides, it is slightly confusing that you use both P and $P/N(D, 1)$ knowing that in fact $N(D, 1)$ is fixed. Would it not be clearer to use the same x axis for these three pictures ?

Response to reviewer comments on "Spectral Bias and Task-Model Alignment Explain Generalization in Kernel Regression and Infinitely Wide Neural Networks"

Abdulkadir Canatar, Blake Bordelon and Cengiz Pehlevan

We thank the reviewers for their encouraging, insightful and constructive comments and suggestions. We have revised our manuscript to address all the concerns, and believe that it has substantially improved. The most important changes to our manuscript are as follows:

- Our setup is extended account for a more general class of target functions, those that are not expressible by the kernel. Now, sections SI.1 and SI.2 discuss the case where target function is of the form $\bar{f}(\mathbf{x}) = \sum_{\rho=1}^N \bar{w}_{\rho} \sqrt{\eta_{\rho}} \phi_{\rho}(\mathbf{x}) + \sum_{\rho=N+1}^{\infty} a_{\rho} \phi_{\rho}(\mathbf{x})$ where $\{\eta_{\rho>N} = 0\}$ for the kernel. We extended the generalization error formula in SI.53 to account for this case.
- We calculated the expected estimator and its variance across different training sets using our statistical mechanics setup, and hence obtained the *bias-variance decomposition* of the generalization error. We showed that the non-monotonicity in the generalization error is solely caused by the variance of the estimator. We added new figures (new Fig. 4B and C) in the main text to demonstrate this fact, applied our results to the real data experiments in the updated Fig. 2 and extended our theoretical calculations in Section SI.2.
- We compared the average estimator to the experimental results for one dimensional data and found excellent agreement with theory (new Fig. 4A).
- We calculated training error using our replica setup (Section SI.2) and obtained excellent agreement with experiment with Gaussian RBF regression on real data (new Fig. SI.1).
- We extended our discussion to include relevant work, their connections to our theory and the current limitations of our theory. We furthermore included a Supplementary Discussion section in Supplementary Information to further discuss our theory's connection to previous literature in detail.
- We extended the details of numerical experiments section in Supplementary Information substantially to include more information about each experiment.
- All codes for the experiments and the figures are uploaded to the following Github Repository: <https://github.com/Pehlevan-Group/kernel-generalization>

We respond in detail to individual points below; referee comments are set in *violet*, and our replies in black.

NOTE: When we quote text from our submission, we changed the numbering of references to match the numbering in this document to make our rebuttal as self-contained as possible.

1 Reviewer #1

1.1 Main contributions

This paper presents an analytical derivation of the generalization performance of kernel regression in a very general setup (applicable to any kernel and any data distribution). The authors clearly show how one can predict performance by studying the interaction between the inductive bias of the kernel (i.e. its eigendecomposition in a basis of orthogonal functions) and the task at play (i.e. the decomposition of the data distribution in this basis).

The authors first show results on classic vision datasets (MNIST and CIFAR10), giving intuition on the kernel-task alignment, then turn to two more theoretical setups. First, the band-limited kernel RKHS setup, which includes linear regression, in which case they obtain a simple formula for the test error enabling a detailed study of the presence of a non-monotonicity in the test error in the (λ, σ) phase-space. Second, the rotationally-invariant setup, where the degeneracy of the eigenmodes enables to reuse results from the band-limited setup in multiple "learning episodes", corresponding the different degenerate eigenmodes.

The mathematical derivation of the generalization error is sound and elegant, albeit not fully rigorous in two ways : (i) it relies on a gaussian approximation, and (ii) it uses the Replica Method from statistical physics. However, I believe the strength of the expression obtained, and the excellent agreement with numerics fully justify the use of such methods.

Overall, the text is very well written, the figures are extremely clear, and the results are significantly more general than previous efforts. The authors put significant emphasis on making the results understandable: the phase spaces of figs 3 and 4 are excellent, and the effective noise and regularization of eq. 9 give a very intuitive understanding of the multiple descent curve (which was rather obscurely presented in previous work). For all these reasons, I think this paper is of strong interest to the community.

We are grateful for the reviewer's careful assessment of our work and encouragement. We hope that our revisions have addressed the reviewer's concerns which we respond below.

1.2 Literature review

In my opinion, the literature review needs to be substantially extended. The paper needs a clearer discussion of other works discussing multiple non-monotonicities, which in the current state are cited without much details [1,2,3], or missed out [4,5,6]. In particular, [1], which obtains a very similar multiple descent curve (in a more limited setup), deserves more discussion. [6] is the most related paper studying kernel regression on real data (but only with random Fourier features) and crucially needs to be cited. I would separate these references into those with several parameter-wise nonmonotonicities [5] and those with several sample-wise nonmonotonicities [1,2,3,4]. It would also be helpful to put the sentence appearing in the conclusion – "Our results extend the finding on the former type of peak, demonstrating the possible existence of many error peaks at each $P = O(D^K)$. The non-monotonicity we observed in the white band-limited RKHS is due to a peak of the latter type" in the introduction. Here, the terminology of linear and nonlinear peak introduced in [3] could be helpful.

- [1] Liang et al. 2020, Multiple Descent of minimum-norm interpolants [1]
- [2] Nakkiran et al. 2020, Optimal Regularization can mitigate double descent [2]
- [3] d'Ascoli et al. 2020, Triple Descent and the two kinds of overfitting [3]
- [4] Chen et al. 2020, Design your own generalization curve [4]
- [5] Pennington et al. 2020, The Neural Tangent Kernel in high dimensions [5]

[6] Couillet et al. 2020, A random matrix analysis of random Fourier features [6]

We agree with the reviewer that our discussion was not sufficient and we are grateful for the recommendations. In the current version, we significantly expanded our discussion and main text, providing citations and making connections to the works listed above and others. We also re-organized our discussion as suggested by the reviewer to discuss model-wise and sample-wise non-monotonicities separately. We copy the relevant changes and the discussion here for reference.

- Regarding Liang et al. 2020, we commented on the similarity of their result on having multiple descents at $P \sim D^K$ starting at line 329:

“When all stages are considered, it is possible to observe learning curves with *multiple descents* with at most one peak per stage. The presence and size of the descent peak depends on the level of noise in the data and the effective regularization as shown in Eq. (8) and Eq. (9). Similar observations of multiple peaks in the learning curves were made in [7] in the context of ridgeless regression on polynomial kernels.”

- Regarding Nakkiran et al. 2020, we established the connection between their optimal regularizer for linear regression and our result starting at line 262:

“Based on this explicit formula, one could choose a large enough λ to mitigate the peak to avoid overfitting for a given noise level (Fig. 3D). However, larger λ may imply slower learning (See Fig. 3B and SI) requiring more training samples. By inspecting the derivative $\frac{\partial E_g}{\partial \lambda} = 0$, we find that $\lambda^* = \sigma^2$ (yellow dashed line in Fig. 3D) is the optimal choice for ridge parameter, minimizing $E_g(\alpha)$ for a given σ^2 at all α (Fig. 3C). For $\lambda > \lambda^*$ the noise-free error term increases from the optimum whereas $\lambda < \lambda^*$ gives a larger noise term (SI). Our result agrees with a similar observation regarding the existence of an optimal ridge parameter in linear regression [2].”

and starting at line 325:

“Similar to the white band limited case, optimal regularization happens when

$$\tilde{\lambda}_L = \tilde{\sigma}_L^2,$$

minimizing $E_g^{(L)}(\alpha)$ for a given $\tilde{\sigma}_L$ for all α . This result extends the previous findings on linear regression [2] to the large class of rotation invariant kernels.”

- Rest of the recommended work has been covered in the discussion section. We copy here the relevant portion:

“ ...

One goal of our present work is to provide a framework that incorporates structural information about the data distribution into a realistic prediction of generalization performance that holds for real data and any kernel. Indeed, a recent study suggested that structure in data allows kernel methods to outperform pessimistic generalization expectations based on the high ambient dimension [8]. In a different setting, authors of [6] calculate test error of random Fourier features model using random matrix theory techniques without strong assumptions on data distribution and obtain excellent agreement on real datasets. Overall, our results demonstrate how data

and inductive biases of a model interact to shape generalization behavior, and in particular the importance of the compatibility of a learning task with the model for sample-efficient learning. Our findings elucidate three heuristic principles for generalization in kernel regression.

...

The third phenomenon we explore is how non-monotonicity can appear in the learning curves when either labels are noisy, or the target function has modes that is not expressible with the kernel. In the analytically tractable models we considered, this is related to a phase transition related to variance explosion in the bias-variance decomposition of the generalization error, appearing in separate learning stages for the rotation invariant kernels. Non-monotonicity can be mitigated with explicit or implicit regularization [9, 10, 11]. We show the existence of an optimal regularizer, independent of sample size, for our theoretical settings. When applied to linear regression, our optimal regularizer matches that previously given by [2].

Non-monotonicity in generalization error gathered a lot of interest recently. Many studies pointed to absence of overfitting in overparameterized machine learning models, signaled by a peak and a subsequent descent in generalization error as the model complexity, or the number of parameters, increases, and the model transitions from an underparameterized to overparameterized (interpolating) regime [12, 13, 14, 15, 9, 10, 3, 16, 17, 6]. Multiple peaks are also possible in this context [5]. Our work provides an explanation for the lack of overfitting in overparameterized models by elucidating strong inductive biases of kernel regression, valid even in the interpolation limit, which includes infinitely overparameterized limits of neural networks. Non-monotonicity as a function of sample size has also been observed previously in many models [13, 18, 19, 15, 11], including ones that show multiple peaks [2, 3, 7, 4]. A closely related study obtained an upper bound for test risk in ridgeless regression which shows non-monotonic behavior with increasing sample size whenever $P \sim \mathcal{O}(D^L)$, consistent with our results on rotation invariant kernels and isotropic data.

An interesting comparison can be made between the multiple peaks we observed in our analytically solvable models and the multiple peaks observed in random features models [3, 5]. In these models, one of the peaks (termed "nonar" in [3]) happens when the number of samples reaches the number of features, and thus the number of parameters of the model, crossing the interpolation threshold. While the peak we observed in the white band limited case with nonar features also happens at the interpolation threshold ($P = N$), the mechanisms causing double descent are different. In random features models, this peak is due to variance in the initialization of the random feature vectors. In our example, such variance does not exist. The peak is due to overfitting the noisy labels, and disappears when there is no noise. The peaks observed for the rotationally invariant case has a more subtle connection. In each learning stage, peaks occur when number of samples reach the degeneracy of eigenfunctions in that stage. While kernel regression is non-parametric, one can think of this again as crossing an interpolation threshold defined by the dimensionality of the feature space in the large D limit. Like the white band limited case, these peaks are due to overfitting noise.

... "

Furthermore, we also introduced the following sentence in the abstract according to the reviewer’s suggestion:

"We find that multiple learning stages exist, one for each scaling of the number of training samples with an integer power of the input dimension and show that each stage might present non-monotonicity in generalization error."

1.3 Comments

I had a hard time understanding was whether the results are valid at finite size, both for D and for P . For D : I thought eq. 9 was valid only for infinite D but in fig. SI.3 the authors seem to say there is a finite and infinite version of the formula (the dashed lines of panels A and B respectively. For P : the fact that P appears for the scaling makes things confusing. If the formula is valid for infinite P , how come it works so well even for P of order 10 in figs 2 and 5 ? It would be nice to have a clearer discussion on the domain of validity of each formula. In fact, more emphasis could be put on the fact that the domain of validity seems to range down to very small values $P=10$, which is very surprising ! How many runs are averaged to obtain such neat curves ?

We thank the reviewer for these questions and incentivizing us to clarify the range of validity of our results. We agree that these points were not well-explained.

The generalization error formula given in Equation 4, which we copy here,

$$E_g = \frac{1}{1-\gamma} \sum_{\rho} \frac{\eta_{\rho}}{(\kappa + P\eta_{\rho})^2} (\kappa^2 \bar{w}_{\rho}^2 + \sigma^2 P\eta_{\rho}),$$

$$\kappa = \lambda + \sum_{\rho} \frac{\kappa\eta_{\rho}}{\kappa + P\eta_{\rho}}, \quad \gamma = \sum_{\rho} \frac{P\eta_{\rho}^2}{(\kappa + P\eta_{\rho})^2}. \quad (4)$$

arises from the saddle point of the partition function. We find that to define a useful thermodynamic limit in which the saddle point is exponentially dominant so that our Equation 4 is exact, the limits we need to take differ depending on the dataset and the kernel. These limits entail taking $P \rightarrow \infty$ (Eq.s SI.30 and SI.31) limit and scaling other quantities with P which in turn govern how kernel eigenvalues scale. This is the reason P appears explicitly above multiplying η_{ρ} . We provide two analytically solvable examples:

1. White-band limited kernel: We find that the natural scaling is to take $P \rightarrow \infty$ and $N \rightarrow \infty$ with $\alpha = P/N \sim \mathcal{O}(1)$, and $D \sim \mathcal{O}(1)$ (or $D = N \sim \mathcal{O}(P)$ in the linear regression case), and $\eta_{\rho} = \frac{1}{N}$.
2. Rotation Invariant Kernels: There are multiple $P, D \rightarrow \infty$ limits leading to different learning stages. We consider a separate limit for each degenerate eigenvalue L while keeping P/D^L finite. Eigenvalues scale with D , $\eta_K \sim \mathcal{O}_D(D^{-K})$.

The corresponding “infinite- P ” limits of Equation 4 are given in Equations 7 and 9 of the main text respectively.

We *empirically* observe that Equation 4 describes low- P values very well. It is for this reason, we chose to express Equation 4 the way it is, before taking a infinite- P limit in the context of a particular kernel and data distribution. We agree that Equation 4 did not have to work so well for low- P .

In kernel regression figures, we typically average over about 100 simulations to plot our generalization error curves. We now also plot the standard deviation to show the variability in the curves. Standard deviation drops rapidly with P , making our theory a very good descriptor for a typical case, as expected from statistical mechanics arguments. However, for low- P , there is considerable variation due to the stochastic nature of sampling the training dataset, again as expected. We provide a copy of the (new) Figure 5 here:

Figure R.1: Figure 5 in main text. Note the large standard deviations when P is small. In (A), the dashed lines are theory, calculated using Eq. 9. Full caption: (A) Phase diagram for non-monotonic learning curves obtained from the theory by counting the zeros of $\frac{\partial E_g}{\partial \alpha}$. Colored squares and colored circles correspond to curves in panel (C) and panel (D), respectively. (B) Kernel regression with Gaussian RBF $K(\mathbf{x}, \mathbf{x}') = e^{-\frac{1}{2D\omega^2}\|\mathbf{x}-\mathbf{x}'\|^2}$ with $\omega = 3$, $D = 100$ and noise-free labels. Target is $\tilde{f}(\mathbf{x}) = \sum_{k,m} \bar{w}_{km} \sqrt{\eta_{km}} Y_{km}(\mathbf{x})$ with random and centered weights \bar{w}_{km} such that $\langle \bar{w}_{km}^2 \rangle = \eta_{km}$ (see SI.5.2.2). Dashed lines represent the locations of $N(D,1)$ and $N(D,2)$, showing different learning stages. (C, D) Generalization error for Gaussian RBF kernel for various kernel widths ω corresponding to specific $\tilde{\lambda}_L$'s and noise variances $\tilde{\sigma}_L$ pointed in the phase diagram in $D = 100$. Solid lines - theory (Eq. 4). Larger regularization suppresses the descent peaks, which occur at $P^* \sim N(D,L)$ shown by the vertical dashed lines. (C) Varying $\tilde{\lambda}_L$ with fixed the $\tilde{\sigma}_L$. (D) vice versa. For fixed noise, we observe an optimal $\tilde{\lambda}_1$ for up to $P/N(D,1) \sim 10$ after which the next learning stage starts. Error bars indicate standard deviation over 300 trials for (B) and 100 trials for (C, D)

Finally, Fig. SI.3 (currently Fig. SI.5) originally has been included to demonstrate some of these points in the case of rotationally invariant kernels. In panel (A), we present perfect agreement with a kernel regression task and Equation 4. In panel (B), we show how infinite- P, D limits, or Equation (9) with $\alpha = P/N(D, L) \sim \mathcal{O}(1)$ for $L = 1, 2$. This figure shows the learning stages and validity domains of the infinite- P, D expressions given in Eq. 9.

In order to clarify these points further, we performed the following changes to the paper:

1. In figure captions, we included which equation is being plotted as theory curves.
2. In order to explain further the figures Fig. SI.5 and Fig. SI.6 (previously Fig. SI3 and Fig.

SI.4), we added the following description above Fig. SI.6 starting line 1437:

“In each panel, the label noise variance chosen according to $g(\tilde{\lambda}_k)$ corresponding to the learning stage $k = 1, 2, 3$. When $\tilde{\sigma}_k^2 < g(\tilde{\lambda}_k)$, generalization error falls monotonically at the learning stage k and when $\sigma^2 > g(\tilde{\lambda}_k)$ we observe double-descent. Fig. SI.6B, on the other hand, demonstrates how infinite P and D limit of E_g , Eq. (9), predicts the regression experiments at each learning stage (corresponding to different panels) when P is finite. We observe that a later learning stage starts before the previous learning stage saturates to its asymptotic value.”

3. We extended the discussion below Eq.(4) in the main text starting line 127 to clarify the validity of our results and point to the finite- P agreement. We copy the relevant text here:

“Formally, this equation describes the typical behavior of kernel regression in a thermodynamic limit that involves taking P to infinity. In this limit, variations in kernel regression’s performance due to the differences in how the training set is formed, which is assumed to be a stochastic process, become negligible. The precise nature of the limit depends on the kernel and the data distribution. In Section 3, for two different analytically solvable cases, we identify natural scalings of N and D with P , which in turn govern how the kernel eigenvalues η_ρ scale inversely with P . We further give the infinite- P limits of Eq. (4) explicitly for these cases. In practice, however, we find that our generalization error formula works very well for even finite P . Therefore, we use Eq. (4) to predict the average behavior of kernel regression for as low as a few samples. We observe that the variance in learning curves due to stochastic sampling of the training set is significant for low- P , but decays with increasing P ...”

4. To improve our technical presentation, we introduced the following text below Eq. SI.27, starting on line 946,

“The reader may question the dependence of various quantities in S on P , since we are taking a $P \rightarrow \infty$ limit. This is because we want to keep our treatment general. Depending on the kernel and data distribution, there are other quantities here that can scale with P . Specific examples will be given.”

and the following text below Eq. SI.44, starting on line 1042

“Note that at this point we have already taken the $P \rightarrow \infty$ limit. P still appears in these expressions to consider different scaling limits for kernel eigenvalues.”

5. In the caption of each figure, we included how many experiments the figures are averaged over. We now show the standard deviation in these curves, giving the reader a sense of the variance in different experiments. The standard deviation falls rapidly with P in all figures.
6. We uploaded all our code to a GitHub repository linked in the paper. Interested readers can reproduce our results easily.

More discussion could be given on the double descent observed for CIFAR10 experiment : what determines the location of the peaks and why are they shifted as noise increases ?

We thank the reviewer for this fair question. Non-monotonicity results from a complex combination many effects including noise, the choice of kernel, data dimensionality and its distribution.

Unfortunately we do not have a clear answer to what determines the location of the peaks and why are they shifted as noise increases for CIFAR-10, except for the fact that the behavior is well-explained by our formula as plotted in the figures. In particular, we have not found an interpretable break-down of our analytical expressions that gives further insight in this case, nor analogues of the “effective noise” and “effective regularization” applicable here. On the other hand, we have fully specified the non-monotonic behavior in our analytically solvable models, e.g. rotationally invariant kernels and spherical data in the infinite P, D limit. We added the following sentence to our discussion to make this point starting at line 382:

“.. We showed that our expression explains generalization on real datasets, and provided a detailed analysis of its application to band-limited kernels with white spectra and the widely used class of rotation invariant kernels [20, 21] operating on spherical data. For the latter case, we defined an effective regularization and an effective noise which govern the generalization behavior, including non-monotonicity of learning curves. It will be interesting to see if analogues of these concepts can be defined for real datasets.”

However, we took the reviewer’s comment as an opportunity to provide some more insight to the non-monotonicity through a new calculation. Using our method, we calculated the bias-variance decomposition of the generalization error. We showed that bias generically decreases with the number of samples, and non-monotonicity is solely caused by the variance. We plot bias and variance for CIFAR10 and MNIST and show that non-monotonicity is due to variance. These results are in new panels Fig. 2D, E, copied here in Figure R.2 for convenience. Along the way, we also calculated the average estimator. All these calculations constitute a major expansion of our results and led to new sections in the supplements, new panels and figures in the main text. We thank the reviewer once again for providing this opportunity to strengthen our paper!

In particular,

- The supplementary information is significantly expanded with the new calculations. Please refer to Section SI.2 and especially Section SI.2.2.3. The following text is added to the main text starting at line 158 to summarize these results:

“Generalization error can exhibit non-monotonicity which can be understood through the bias and variance decomposition [9, 10, 3], $E_g = B+V$, where $B = \int d\mathbf{x} p(\mathbf{x}) \left(\langle f^*(\mathbf{x}) \rangle_{\mathcal{D}} - \bar{f}(\mathbf{x}) \right)^2$ and $V = \left\langle \int d\mathbf{x} p(\mathbf{x}) \left(f^*(\mathbf{x}) - \langle f^*(\mathbf{x}) \rangle_{\mathcal{D}} \right)^2 \right\rangle_{\mathcal{D}}$. We found that the average estimator is given by $\langle f^*(\mathbf{x}; P) \rangle_{\mathcal{D}} = \sum_{\rho} \frac{P\eta_{\rho}}{P\eta_{\rho} + \kappa} \bar{w}_{\rho} \psi_{\rho}(\mathbf{x})$, which monotonically approaches the target function as P increases, giving rise to a monotonically decreasing bias (Methods and SI). However, the variance term arising from the variance of the estimator over possible sampled datasets \mathcal{D} is potentially non-monotonic as the dataset size increases. Therefore, the total generalization error can exhibit local maxima.”

- In the updated Fig. 2, we show experimentally that the double-descent in CIFAR10 and MNIST occur due to the variance of the estimator. We copy the figure here for convenience, see Figure R.2.
- We added a new figure, Figure 4, to demonstrate our new theoretical results for bias-variance decomposition. We copy the figure here for convenience, see Figure R.3.

Figure R.2: Figure 2 in main text. Kernel regression with Gaussian RBF $K(\mathbf{x}, \mathbf{x}') = e^{-\frac{1}{2D\omega^2}\|\mathbf{x}-\mathbf{x}'\|^2}$ for MNIST and CIFAR-10 with kernel bandwidth $\omega = 0.1$, ridge parameter $\lambda = 0.01$. (A) Generalization error $E_g(p)$ when $\sigma^2 = 0$: Solid lines are theory (Eq. 4), dots are experiments. (B) Kernel eigenvalues and (C) cumulative powers for MNIST and CIFAR-10. (D, E) Generalization error when $\sigma^2 = 0.5$ with its bias-variance decomposition for MNIST and CIFAR-10 datasets, respectively. Solid lines are theory, markers are experiments. Error bars represent standard deviation over 160 trials. Bias and variance are obtained by calculating the mean and variance of the estimator over 150 trials, respectively.

Figure R.3: Figure 4 in main text. Bias-variance decomposition of generalization error for the white band-limited example. (A) Average estimator for kernel regression with $K(x, x') = \sum_{k=1}^N \cos(k(x - x'))$ on target function $\bar{f}(x) = e^{4(\cos x - 1)}$ with mean subtracted for different values of $\alpha = P/N$ when $\lambda = \sigma^2 = 0$. Estimator early approaches to the target function and estimates it perfectly when $\alpha = 1$. Dashed lines are theory. (B) With the same setting in Fig. ??, when $\lambda = 0$ and $\sigma^2 = 0$, the bias is a monotonically decreasing function of α while variance has a peak at $\alpha = 1/2$ yet it does not diverge. (C) When $\lambda = 0$ and $\sigma^2 = 0.2$, we observe that the divergence of E_g is only due to the diverging variance of the estimator. In (B) and (C), solid lines are theory, dots are experiments. Error bars represent the standard deviation over 15 trials.

One slight concern is in Figure 5B. For small values of P , the experimental and theoretical points do not match very well when the variance becomes large, despite the network operating in the over-parameterized regime. Even though the error bars of the numerical curve seem to just about reach the analytical curve (not clear at very low P), they are systematically biased, signalling that there is a non-captured phenomenon is at play. Is this simply a finite-size effect, or something more profound ?

Excellent point! There is certainly a bias in low- P regime where the kernel regression and theory matches perfectly while the neural network experiments are significantly biased to higher risks. Note that our theory only explains kernel regression and relate to neural networks only indirectly via the NTK regression correspondence in large widths. Hence reviewer's point about low- P phenomena remains to be understood with a systematic study of finite-width neural networks and their kernel correspondence. This is beyond the scope of our work.

However, inspired by the works [22, 10, 13], we utilized neural network ensembling to see how it will affect the low- P deviation. Recall that neural network ensembling mitigates double-descent in the high- P region since it reduces the effect of initialization variance [10] and brings the neural network test risk closer to the kernel prediction. Our empirical observations suggest that ensembling also helps reducing the discrepancy between NTK regression and finite-width neural network test risks as shown in Fig.R.4. For ensembling of size 10, we obtain almost perfect match between kernel regression and neural network experiments in low- P regime. In order to make this point, we updated our discussion under "SI.4.2 Dot-Product Kernels and Neural Tangent Kernel" and added a new figure (Fig.SI.9) in SI with an explanation of how ensembling neural networks brings finite-width neural networks closer to their NTK analogue. We added the following paragraph under the section SI.4.2 starting line 1475:

"We further observe a discrepancy between NTK regression and neural network training at low- P where the generalization error is systematically biased towards higher values than its kernel regression counterpart. While this effect requires a further study of the correspondence between finite width neural networks and NTK regression, and remains to be fully understood, we empirically observe that the method of neural network ensembling [22, 10, 13] removes this discrepancy (Fig. SI.9B). Neural network ensembling is known to reduce the variance due to random parameter initializations [10] and hence also prevents the high- P mismatch we observe in Fig. SI.9A."

1.4 Minor Comments

There are a few notational mixups.

We greatly appreciate for pointing out the mistakes/typos in the text. The relevant changes has been done in the text and mixups are explained below.

- *In Methods / Diagonalizing the kernel on real datasets : N is the size of the dataset instead of P .*

Thank you! We had used N to denote the size of whole dataset (training + test) on which we calculate the eigenvectors and eigenvalues. P only denoted the size of training set. To avoid confusion, we now use M for the size of whole dataset, P for the size of the training set and N for the number of features/eigenfunctions.

- *SI : M seems to denote what is called N in main text.*

Figure R.4: Effect of ensembling on neural network learning curves. The crosses denote the neural network experiments. We find that ensembling leads better matching between the neural network and kernel regression experiments. The effect is most visible in the blue curve. Note that this figure has more simulations than Fig.SI.9.

We are thankful for pointing out this typo. Throughout the SI, letter M is replaced by letter N and it denotes the number of features (also non-zero eigenvalues) of the kernel.

- *Conclusion* : “the number of parameters P reach...”. This typo urgently needs correction as it can be very misleading !

Thank you for correcting this crucial mistake. The sentence has been changed to “when number of samples, P , reach the number of input dimensions, D ...”

- *Fig 2* : x axis should be P

Typo has been fixed.

- *Fig 3* : x axis is α

Typo has been fixed.

- *Fig SI.3* : missing the vertical lines

Vertical lines are restored.

- *Line 54* : “generalisation in”

Typo has been fixed.

- *Fig SI.4* : why only $D=20$? Can't we make D bigger to have less finite-size effects ? Or plot for increasing D to see the gaps “fill in” ?

Thank you very much for this suggestion. Note that in numerical experiments we cannot choose very large dimensions since the number of samples required to learn a training set

with sample dimension D scales as $P \sim D^k$. Hence to distinguish between at least two modes we need D^2 samples and computationally we can only access sample sizes of $P \approx 10,000$. Therefore, $D = 20$ gives us three learning stages, while $D = 100$ gives only two. We present the experiments done with larger D in Fig.R.5 where data is averaged over 30 trials and errorbars are removed for clearer display. We observe that the transition from 1st learning stage to second one occurs later for larger input dimensions (e.g. $P_{1 \rightarrow 2} \approx 2 \times 10^2$ for $D = 20$, $P_{1 \rightarrow 2} \approx 10^3$ for $D = 100$ and $P_{1 \rightarrow 2} > 10^5$ for $D = 1000$) but the gaps do not completely fill in since the input dimension is still pretty small.

Figure R.5: Asymptotic learning curves vs. the finite dimensional regression experiments for $D = 20, 100, 1000$.

2 Reviewer #2 (Remarks to the Author)

2.1 Summary of the paper

This paper provides a theoretical formulation, using methods of statistical physics, of generalization error in kernel regression and infinite-width neural networks.

Outlined contribution of the work from the authors are:

[C1] Analytic expression for generalization error applicable to any kernel or data distribution (Eq 4): from using techniques from statistical physics

[C2] Application to this theory to ‘real’ and ‘synthetic’ datasets for various kernels. Which covers the infinite-width limit of deep neural networks.

[C3] Inductive bias of kernel regression towards ‘simple functions’ characterized by spectral properties of the kernel

[C4] Show more data could hurt generalization if noisy, leading to non-monotonic learning curves.

[C5] Study broad class of rotation invariant kernels, on spherical uniform data and large input dimensions which allow detailed mathematical analysis.

2.2 Significance

Understanding generalization of overparameterized models in machine learning is a fundamental question which is an active research area. Traditional uniform convergence bounds often does not provide significant insight into how modern overparameterized models generalized well to unseen data. Therefore new techniques to understand “typical” generalization property instead of “worst” case is becoming increasingly more important. This paper uses techniques from statistical physics (replica method) allowing such analysis.

I think this work can be interesting to the general scientific community 1) since machine learning and AI techniques are becoming ubiquitous in the field of science 2) this work addresses important “scientific questions” for important complex learning systems.

I believe provided theory is an important step towards understanding generalization in machine learning and in particular overparameterized neural networks. Experiments in this paper showing various generalization phenomena demonstrate one could obtain insights on inner workings of how learning is happening in kernel type models.

Given that, there are some limitations I see (also detailed more in [Technical concerns] below).

We thank the reviewer for his/her constructive and detailed comments and suggestions. They improved our paper significantly. Below we provide our responses. We believe we addressed all the concerns.

First the claim that “our theory can predict which kernels or neural architectures are well suited” seems a little too strong as what proposed theory actually could do. If my understanding regarding the target function in [technical concerns] is correct, analytic expression for generalization is limited to very specific target functions, and unclear whether it applies to natural labels that exist in the real dataset.

Thank you for your comments and raising these important concerns. Contrary to the reviewer’s statement, our methods are applicable to any smooth (finite trace) kernel and any square-integrable target function, which include the natural labels appearing in real datasets represented in a one-hot vector form. We apologize for not being clear in our presentation. We provide a detailed response and actions we have taken to clarify our presentation under technical concerns.

We do want to point here that in our manuscript we already demonstrated the applicability of the theory to real datasets (MNIST and CIFAR-10) by several experiments in Figures 1 and 2. For these examples, target functions are the original digit/object label assignments to data represented as one-hot vectors. We already explain in Methods, in detail, our procedure for doing this by diagonalizing the kernel on real data and projecting the label vectors to kernel eigenvectors. We added the following text in the main paper (starting line 173) to clarify:

“In our first experiment, we test our theory using a 2-layer NTK [23, 24] on two different tasks: discriminating between 0s and 1s, and between 8s and 9s from MNIST dataset [25]. We formulate each of these tasks as a kernel regression problem by considering a vector target function which takes in digits and outputs one-hot labels. Our kernel regression theory can be applied separately to each element of the target function vector (Methods), and a generalization error can be calculated by adding the error due to each vector component.”

For these reasons, we chose to keep our statement “Consequently, our theory can predict which kernels or neural architectures are well suited to a given task by studying the alignment of top kernel eigenfunctions with the target function for the task.” as is.

Moreover, the current theory requires “eigen decomposition” of the full dataset to obtain kernel eigenvalues and eigenvectors. With similar or even less compute, one could easily do either small k-fold cross validation or leave-one-out cross validation on a given dataset to determine how well a given kernel/neural architecture can perform on a given dataset.

Thank you for pointing this out. We agree that the aforementioned methods are well established model-evaluation techniques however our main purpose is not only evaluating the performance of a kernel machine, but also understanding the theoretical aspects of learning such as implicit regularization, non-monotonicity in generalization error and task-model alignment in an analytical setting. Although empirical methods such as k-fold cross-validation are very useful for evaluating the model performance, they do not give further intuition about the generalization itself. (see [technical concerns] for limitations related to eigendecomposition).

We introduced a new paragraph in Discussion addressing the limitations of our work, which includes the sentence starting at line 449:

“First, the theory requires eigendecomposition of the kernel on the full dataset which is computationally costly.”

Lastly, there are few concerns whether NTK-style kernel learning captures the good generalization property of SOTA finite networks. There are many studies regarding “feature learning” vs “lazy learning” and various finite network training techniques (large learning rate, l2 regularization) that diverts out of vanilla NTK setting. Moreover while many of the analysis is based on rotationally invariant kernels, widely successful convolutional models are not exactly rotationally invariant in the sense defined in this paper. Certainly these are beyond the scope of this paper and would be an interesting follow-up study. I believe, in general, clearly stating current limitations and what future research is required would be service to the research community.

We thank the reviewer for this point, and we agree with everything that the reviewer states. It is correct that our theory can predict neural networks only in the NTK limit of neural networks, and breaks down when the NTK limit breaks down. This is a serious limitation if one wants to apply our theory to finite-sized SOTA neural networks.

To clarify a minor point, within the kernel limit, our theory is applicable to a wide-range of architectures. Recent works have shown that a very general class of neural networks can have a kernel description in the large width limit including feedforward, convolutional and recurrent neural networks [26, 27, 28, 29].

We added the following sentence to the last paragraph of our discussion, which discusses the limitations of our theory, starting at line 450.

“Second, its applicability to state-of-the-art neural networks is limited by the kernel regime of networks, which does not capture many interesting and useful deep learning phenomena [30, 31].”

and this sentence starting line 387

“Our results are directly applicable to infinite-width limit of neural networks that admit a kernel description (including feedforward, convolutional and recurrent neural networks) [24, 32, 26, 33, 27, 28], and explain their inductive bias towards simple functions [34, 35, 14, 31, 36, 37, 38]. ”

2.3 Novelty

- The Replica method is a common technique used widely in the theoretical physics community. Even in the context of machine learning, there were quite a few works utilizing similar techniques to obtain learning curves for kernel based methods (SVM, GP). While there are citations to few of these works, the paper does not clearly state how this submission distinguishes itself.

We thank the reviewer for bringing this point up. Therefore, we added a section new section "SI.7.3 Relation to Other Statistical Physics Approaches to Kernel Machines" under the new Supplementary Discussion in the SI to expand on the previous literature on applications of the replica method to kernel based methods, and the distinguishing features of our work. We would have liked to do this in the main text, however because of word limits, we had to make this compromise.

The related paragraphs read:

"In the statistical physics domain, the replica method has been used to calculate classification learning curves for support vector machines when the data distribution is spherically symmetric and high dimensional, and for a specific class of target functions [39, 40]. These works revealed a countably infinite number of consistent thermodynamic limits: $P, D \rightarrow \infty$ with $P = \alpha D^L$ for integer L , which arise due to the degeneracy of kernel eigenvalues for all spherical harmonics of degree L . In the L -th learning stage, polynomials of degree L are being estimated as α increases. The rotation invariant kernels we discuss in this paper show the same learning stages, though for kernel regression rather than kernel SVM. However, our theory not only applies to spherically symmetric and infinite dimensional settings but is also shown to work for more general data distributions and a wide range of target functions, including realistic datasets such as MNIST and CIFAR. To aid study of kernel methods in these more general distributions, we introduce useful metrics such as cumulative power $C(\rho)$ to quantify the alignment of the kernel with the learning task of interest.

The generalization of regularized linear regression with isotropic Gaussian features was analyzed with the replica method in [41] and diagrammatic methods in [19, 18, 42]. These works predicted learning curves equivalent to a special case of our kernel regression

theory: the white band-limited spectrum. In this special case, we provide a phase diagram showing how effective noise and regularization can alter the non-monotonic behavior of the learning curve. We further show universality of this learning curve and phase diagram to any setting where the kernel admits a truncated Mercer decomposition with all eigenvalues equal. We explicitly illustrate such an equivalence in the spherically symmetric data setting. In the L -th learning stage ($P \approx D^L$ with $D \rightarrow \infty$), any kernel essentially reduces to white band-limited kernel with $N(D, L)$ equal variance modes (spherical harmonics of degree L). We show that the higher degree components of the target act as effective noise while higher degree kernel eigenvalues act as effective regularization, allowing the phase diagram from the white band-limited setting to carry over to this interesting case.

Generalization error for Gaussian process regression, where the target function is a random field with covariance kernel K_t , was analyzed in the average case by Peter Sollich [43, 44]. Using the Sherman-Morrison inverse formula, he derived a continuous approximation of the learning curves from a partial differential equation for κ . In his work, he introduced a variety of approximations to learning curves based on this PDE approach. The learning curves obtained in the present work with the replica method agree with his *lower continuous approximation* in the kernel regression limit. Our theory goes beyond this work both in formalism and the results. Our framework allows computation of other relevant observables (training error, average estimator, bias and variance) by simply including additional sources in our partition function. We provide a detailed analysis of the generalization error expression to elucidate important heuristics for generalization, and further show its applicability to real datasets.

Recent works on the effects of over-parameterization in random feature models has been investigated with the replica method [17, 10] and with random matrix theory tools [9, 15]. In the random features model, the last layer weights in a randomly initialized neural network are trained. In this setting the learning curves depend on the input dimension D , the number of random features N (network width) and the number of data points P . Typically, authors consider fitting noisy linear target functions with these networks. In this model, two types of overfitting peaks can occur, one when $P \approx D$ and one when $P \approx N$ [3]. The relation between these peaks and the ones we see are discussed in the main text. "

- One thing that was not very clearly stated was whether the used method by the author is novel or not. Authors do cite their ICML 2020 publication by the same set of authors (Ref 31) which bulk of the content and contributions is quite similar to this submission. For example, both use a replica method to obtain learning curves of kernel regression and use connection to infinitely-wide neural networks to make statements about overparameterized networks ([C1, C2]). Also similar spectral dependency and exact analysis based on data on the sphere is done ([C3]). To the reviewer, it was not clear whether this work is an "expansion" of Ref 31 for a journal submission or novel orthogonal work. I strongly encourage authors to make this point clear to set the readers' expectations straight.

Thank you for your comments. While our ICML 2020 publication [45] lays the groundwork of this work, the current submission to Nature Communications significantly expands on it both in terms of methodology and results, and brings in many new insight:

- The methods of [45] are based on the continuous approximation method introduced by [44] and solving a coupled PDE equation. The replica method is included in [45] for representing a random matrix inversion in terms of a Gaussian integral while in the current submission the

replica method is used to calculate the finite temperature partition function associated to the energy functional $\mathcal{H}[f]$. Also being a common technique in theoretical physics [46], this method is more general in that one can calculate the expectation value of *any functional/observable* of the learned function $\mathcal{O}[f]$ by calculating:

$$Z = \int \mathcal{D}f e^{-\beta\mathcal{H}[f]+J_1\mathcal{O}_1[f]+J_2\mathcal{O}_2[f]+\dots}$$

$$\langle \mathcal{O}_i[f] \rangle = \frac{\partial}{\partial J_i} \log Z|_{J_i=0} \quad (\text{R.1})$$

For example, we pick $\mathcal{O}[f] = \int d\mathbf{x} (f(\mathbf{x}) - \bar{f}(\mathbf{x}))^2$ to calculate the expected generalization error.

- To demonstrate the generality of this approach further, we added new results on the calculation of the expected training error, the average estimator and its variance in the manuscript by adding appropriate source terms (SI.2). This allowed us to calculate the bias-variance decomposition of the generalization error. We obtained excellent agreement with theory and experiments. Section SI.2 is significantly expanded with these calculations. Main text is also extended. New panels in Figure 2 (D,E) and a new Figure 4 illustrate the results of these efforts. These figures are repeated here in Figures R.2 and R.3. To not repeat ourselves, we kindly ask the reviewer to see our response to the first reviewer’s question about double descent on CIFAR10, starting at the bottom of page 7 of this document, where these new additions are detailed.

One point that is not detailed in our response to reviewer 1 is the new training error calculation, which can be found in SI.2.2.2. We copy here the relevant figure from that section, Figure R.6.

Figure R.6: Kernel regression with Gaussian RBF $K(\mathbf{x}, \mathbf{x}') = e^{-\frac{1}{2D\omega^2} \|\mathbf{x}-\mathbf{x}'\|^2}$ for MNIST with kernel bandwidth $\omega = 0.1$, ridge parameter $\lambda = 0.01$ and noise variance $\sigma^2 = 0, 0.5$. Theoretical predictions for E_g (solid blue line) and E_{tr} (dashed orange line) are in excellent agreement with experiment (blue dots and orange triangles, respectively). Error bars represent standard deviation over 30 trials. (A) When no label noise is present, E_{tr} is always smaller than E_g . (B) When noise is non-zero, E_{tr} may exceed E_g due to explicit presence of label noise in the loss function.

- In this revised manuscript, we provide a stronger result about the spectral bias of kernel regression. In the ICML 2020 paper, we prove that the logarithmic derivatives of mode errors with respect to sample size P are ordered according to the kernel eigenvalues. Here we prove that the “normalized” mode errors E_ρ themselves are exactly ordered for all P : $E_1 \leq E_2 \leq E_2 \dots$. This stronger result allows us to establish that learning tasks with higher cumulative power

distributions $C(\rho)$ achieve lower generalization error. We study the applications on real data and bring up the concept task-model alignment by introducing the cumulative power distribution $C(\rho)$ which characterizes how well the target function is explained by the first k eigenfunctions of the kernel.

- Our previous ICML paper focused solely on the spectral biases of kernel regression and did not explore the influence of noise on generalization. Consequently, it did not identify the possibility of non-monotonicity in the learning curves. In this paper, we study the effect of label noise and analytically characterize the non-monotonicities arising from over-fitting the noisy labels. A significant portion of our results, including phase diagrams, are about non-monotonicity.
- We introduced the white band limited model in this paper.
- Comparisons to CIFAR10 is also new in this paper.

We introduced the following paragraph in Supplementary Discussion, starting line 1707:

“Our recent conference publication on kernel regression and the infinite-width limit of neural networks [45] laid the ground work for the present investigation. However, the current paper significantly expands on it and brings in many new insights. First, the effect of label noise on generalization and the multiple descent phase diagram were not explored in our previous work which we provide an account of here. Second, we provide comparisons to CIFAR-10, not present in the previous work. Third, we introduced and analyzed the white band limited RKHS model. Fourth, we emphasize task-model alignment as a heuristic here and provide a metric for it. Fifth, the field theory formalism presented in this paper is more general than the technique of [45], which only used the replica method to calculate the average of an inverse matrix. The flexibility of the new theory allowed us to compute other observables including training error, the expected estimator, and the covariance of the estimated coefficients over different datasets.”

- In regards to the previous work, I was hoping to find out differences in method/result of [41] (Cohen et al., 2019) since they also use replica methods to obtain learning curves for GP and make connections to infinitely-wide networks. In general, it is hard to tell which contribution in this paper is novel / different from previous work. If spaces are allowed, or at least in the supplement information, I encourage the authors to expand related work and how the result of this paper distinguishes among other works.

Thank you for mentioning this interesting work. Indeed there are similarities between our theory and work by (Cohen et.al 2019) in terms of the methods, however our calculation differs from and is more general than theirs in several ways. We apologize in advance for the technical nature of the discussion below.

- Their initial partition function (G10 in [47]) is same as ours (Eq.15) and both theories require averaging over training dataset (see G11 in [47] vs. $\langle \log Z(J) \rangle_{\mathcal{D}} = \lim_{n \rightarrow 0} \frac{1}{n} (\langle Z(J)^n \rangle_{\mathcal{D}} - 1)$ under Methods section). However, this is a difficult task and their solution around it was averaging it over an ensemble of training sets of size $P \sim Poisson(P)$, called grand canonical partition function, while we average it over only training sets of size P which is more accurate since it does not involve uncertainty of the size of the training set. Note that the standard deviation of their average is \sqrt{P} and their result converges to ours only in the limit $P \rightarrow \infty$.
- After averaging over the “size of the training set”, they needed to average over the training set itself. This requires averaging an exponential over the samples (The expectation value of

the Gaussian under Eq. H1 in [47] and Eq.SI.16 in our paper). We perform this task “exactly” by the Gaussian approximation we introduced under section “SI.2.1 Averaging over Quenched Disorder”, however they need a “small training error” expansion of the exponential in order to proceed. Since we do not perform such an expansion, our calculation corresponds to averaging over infinitely many terms in their power series expansion.

- Their method requires calculating systematic corrections to the estimator which is a tedious process. In fact, [47] calculates the correction only next-to-leading order while our prediction is exact up to the Gaussian approximation and is in fact more general as we showed in new section "SI.2.2.3 Expected Estimator and the Correlation Function".
- To the perturbative order they consider, they find that $\langle f^*(\mathbf{x}; P)^2 \rangle_{\mathcal{D}} = \langle f^*(\mathbf{x}; P) \rangle_{\mathcal{D}}^2$, missing the variance of the estimator which plays a crucial role in our theory for non-monotonicity.
- Their method fails when one considers ridgeless limit $\lambda \rightarrow 0$ and predicts perfect learning even with a single training sample. They resort to renormalization group methods in this limit. On the other hand, our expression yields a well-defined generalization error in this limit, consistent with experiments.
- Besides this methodological differences, our work includes an in depth analysis of various generalization phenomena and heuristics.

To demonstrate that our result is more general, we included an example showing that our theory matches their results when we expand our expression to first order but also includes infinitely many correction terms as a power series in $\kappa - \lambda$ under the section "SI.2.1 Averaging over Quenched Disorder". For reference, we copy the text below. Note that our theoretical expression for the predictor is $\langle f^*(\mathbf{x}; P) \rangle_{\mathcal{D}} = \sum_{\rho} \frac{P\eta_{\rho}}{P\eta_{\rho} + \kappa} \bar{w}_{\rho} \psi_{\rho}(\mathbf{x})$.

"A related result was given in [47] which used a similar technique to calculate the expected estimator. With our notation, their perturbative result for uniform spherical datasets and dot product kernels reads (Eq.(24) in [47]):

$$\langle f^*(\mathbf{x}; P) \rangle_{\mathcal{D}} = \sum_{\rho} \left(\frac{P\eta_{\rho}}{P\eta_{\rho} + \lambda} - \frac{P\eta_{\rho}}{(P\eta_{\rho} + \lambda)^2} \lambda \sum_{\rho} \frac{\eta_{\rho}}{P\eta_{\rho} + \lambda} + \dots \right) \bar{w}_{\rho} \psi_{\rho}(\mathbf{x}), \quad (\text{SI.63})$$

where λ is the ridge parameter and \dots represent the higher order corrections in their perturbative setting. We obtain the same result if we expand our expression for $\langle f^*(\mathbf{x}; P) \rangle_{\mathcal{D}}$ in a power series of $\epsilon = (\kappa - \lambda)$ assuming ϵ is small, and use $\kappa - \lambda = \kappa \sum_{\rho} \frac{\eta_{\rho}}{P\eta_{\rho} + \kappa}$:

$$\frac{P\eta_{\rho}}{P\eta_{\rho} + \kappa} = \frac{P\eta_{\rho}}{P\eta_{\rho} + \lambda} - \frac{P\eta_{\rho}}{(P\eta_{\rho} + \lambda)^2} \lambda \sum_{\rho} \frac{\eta_{\rho}}{P\eta_{\rho} + \lambda} + \dots \quad (\text{SI.64})$$

Hence, our result includes corrections not captured in [47]."

Furthermore, we commented on Cohen et.al. 2019 under the Supplementary Discussion starting at line 1697:

"The equivalence between training infinite-width neural networks and kernel methods, allowed Cohen et al. to study the generalization of wide neural networks with a Gaussian

field theory [47]. They obtained a different expression for theoretical generalization error using a Poisson averaging technique, allowing them to expand their theoretical prediction in powers of $1/P$ at finite λ . We show in equation (SI.64) that their expression for $\langle f^*(\mathbf{x}; P) \rangle_{\mathcal{D}}$ can be obtained from ours by a series expansion in a certain limit, showing that our result contains corrections not captured in their theory. Further, to the perturbative order they consider, they find that $\langle f^*(\mathbf{x}; P)^2 \rangle_{\mathcal{D}} = \langle f^*(\mathbf{x}; P) \rangle_{\mathcal{D}}^2$, missing the variance of the estimator which plays a crucial role in our theory, especially for non-monotonicity. Further, their expressions break down in the ridgeless limit and they resort to a renormalization approach to predict learning curves. Our expression is valid in this limit."

2.4 Technical concerns

- One potential drawback of the theory is that strong assumptions on the target function. Theory always assumes that the target function lies in RKHS and is able to expand in terms of the eigenbasis.

Under some smooth assumption on the labels of the dataset this may be true. However, for a typical image classification task with one-hot labeling, the target function is extremely not smooth and I expect it is either hard to apply theory in this paper or agreement between theory and experiment wouldn't be as good as shown. If I'm not mistaken, the paper should explicitly clarify the limitation, unless it can be misleading.

This is especially true, since the authors boast application to the "real" dataset. Please correct me if I'm mistaken, however, from my reading the test data label always needs to come from "target function \bar{w} "(inferred from training labels) even in the MNIST or CIFAR10 experiments. I believe the actual test set label is quite different from the label assigned by this target function from \bar{w} .

In other words, the theory assumes the target is within the class the models can learn. Often real tasks would have quite complicated task functions, while functions represented by simple kernel or neural architecture are quite far. This will lead to not as good generalization but current setup, to my understanding, does not capture the case.

While I believe understanding the learning curve in this setting is an important problem and can lead to understanding of very important questions of how learning/generalization occurs. There still seems to be quite a gap in understanding real target function.

Thank you for raising these concerns. The target functions in our theory do not need to lie in the RKHS, and our theory is applicable to the typical image classification task with one-hot labeling. In fact, Figures 1 and 2 already show examples of such applications (MNIST and CIFAR-10 with their actual one-hot labels), and the procedure is already explained in Methods. Our theory is indeed very general in terms of its applicability. We apologize for not making these points clear. Below we give a technical discussion.

- Our theory applies to any square integrable target function which does not necessarily lie in the RKHS. The sections SI.1 and SI.2 are updated to explain our reasoning in detail but here we provide a sketch of the main ideas. By Mercer's theorem, the normalized eigenfunctions of the kernel $\{\phi_\rho\}$ form a complete basis for functions which are square integrable on the measure $p(\mathbf{x})$ [21]. Mercer's theorem states:

$$K(\mathbf{x}, \mathbf{x}') = \sum_{\rho=0}^{\infty} \eta_\rho \phi_\rho(\mathbf{x}) \phi_\rho(\mathbf{x}'), \tag{R.2}$$

Since the normalized eigenfunctions $\{\phi_\rho\}$ are complete, the target function can be expanded in this basis

$$f(\mathbf{x}) = \sum_{\rho=0}^{\infty} a_\rho \phi_\rho(\mathbf{x}), \quad a_\rho = \int p(\mathbf{x}) f(\mathbf{x}) \phi_\rho(\mathbf{x}) d\mathbf{x}. \quad (\text{R.3})$$

This operation is always valid provided the target has finite power $\int p(\mathbf{x}) f(\mathbf{x})^2 d\mathbf{x} < \infty$.

However the kernel can have zero eigenvalues $\{\eta_\rho\}$. Defining the null-space indices of the kernel as $\mathcal{N} = \{\rho | \eta_\rho = 0\}$ and its complement as \mathcal{N}^\perp , we note that by the representer theorem [21], the learned function must only depend on those modes with $\eta_k > 0$ since

$$f(\mathbf{x}) = \sum_{i=1}^P \alpha_i K(\mathbf{x}, \mathbf{x}_i) = \sum_{i,\rho} \alpha_i \eta_\rho \phi_\rho(\mathbf{x}_i) \phi_\rho(\mathbf{x}) = \sum_{\rho \notin \mathcal{N}} w_\rho \psi_\rho(\mathbf{x}), \quad (\text{R.4})$$

where $w_\rho = a_\rho \eta_\rho^{-1/2}$ and $\psi_\rho = \sqrt{\eta_\rho} \phi_\rho$. We can express the target function as:

$$\bar{f}(\mathbf{x}) = \sum_{\rho \notin \mathcal{N}} \bar{w}_\rho \psi_\rho(\mathbf{x}) + \sum_{\rho \in \mathcal{N}} \bar{a}_\rho \phi_\rho(\mathbf{x}) \quad (\text{R.5})$$

where $\bar{w}_\rho = \bar{a}_\rho \eta_\rho^{-1/2}$. Therefore the generalization error is

$$E_g = \left\langle \left(\sum_{\rho \notin \mathcal{N}} (w_\rho - \bar{w}_\rho) \psi_\rho(\mathbf{x}) - \sum_{\rho \in \mathcal{N}} \bar{a}_\rho \phi_\rho(\mathbf{x}) \right)^2 \right\rangle = \sum_{\rho \notin \mathcal{N}} \eta_\rho (w_\rho - \bar{w}_\rho)^2 + \sum_{\rho \in \mathcal{N}} \bar{a}_\rho^2 \quad (\text{R.6})$$

The estimator is only able to learn w_ρ for $\rho \notin \mathcal{N}$, and reduce generalization error in these modes. The remaining modes are not learnable, and lead to a residual error $\sum_{\rho \in \mathcal{N}} \bar{a}_\rho^2$. Our replica analysis calculates the error due to the learnable modes. All sums in the replica calculation therefore only run over modes where $\eta_\rho > 0$. The total error, however, can be calculated by adding back the residual error.

Furthermore, we showed that the residual error $\sum_{\rho \in \mathcal{N}} \bar{a}_\rho^2$ acts as an implicit noise on the target labels even when the noise variance $\sigma^2 = 0$ (See the discussion below Eq. SI.53). Consequently, for target functions which are out-of-RKHS, we demonstrated the existence of double-descent even when there is no label noise. We show this and the irreducible error in the learning curve in a new Fig. SI.2 in the Supplementary Information (here Fig. R.7):

- For real datasets, we define the input space \mathcal{X} as the set of points in the training and test sets (as many samples are available, M) so $\mathcal{X} = \{\mathbf{x}_1, \mathbf{x}_2, \dots, \mathbf{x}_M\}$. The distribution is simply $p(\mathbf{x}) = \frac{1}{M} \sum_{i=1}^M \delta(\mathbf{x} - \mathbf{x}_i)$. In this setting there are exactly M eigenfunctions (eigenvectors) and all possible target functions $\mathbf{y} \in \mathbb{R}^M$ can be realized on these M points by taking linear combinations of these eigenfunctions, including one-hot-vectors. Because one-hot-vectors have finite norm in this finite dimensional space, our theory is applicable to such targets. We use the *actual training labels in one-hot-vector form* as the target functions on \mathcal{X} . The target weights \bar{w} are computed by projecting the one-hot actual training labels onto the eigenvectors found by diagonalizing the empirical kernel Gram matrix against the distribution $p(\mathbf{x})$. This is explained in Methods. We expanded our explanation of this point in the main text, quoted above under our response to the related question in “2.2 Significance”.

Figure R.7: Generalization of kernel regression on a noise free target function with components outside of RKHS can still be estimated with our theory. The outside of RKHS components generate an irreducible error $E_g(\infty)$ and act as effective noise.

- The authors obtain a solution assuming replica symmetry. Most cases, this is a fine assumption and often a good interesting starting point of analysis. This can be also retrospectively validated by empirical check that assumption was good. So I believe this is not a significant issue but I do wonder if the authors have considered the possibility of replica symmetry breaking (RSB) solutions. In statistical physics, this often leads to more interesting phenomenology and wonder in the context of kernel methods or infinitely wide neural networks, whether one wants/needs to consider RSB solutions to capture a more realistic setting.

Thank you for pointing out this aspect of the theory. RSB ansatz is often useful for non-convex optimization tasks and reveal interesting physics about spin glass phases as well as machine learning applications. We found that the RS ansatz matches experiment perfectly, and therefore did not pursue RSB. Finally, although it seems not peer reviewed yet, a very recent preprint (approximately a week old at the date this is written) claims to give a rigorous proof of our result [48], however, the proof seems to depend on a conjecture similar in spirit to our Gaussianity approximation.

We added the following sentence to our discussion in line 452:

"Third, our theory uses a Gaussian approximation and a replica symmetric ansatz. While these assumptions were validated by the remarkable fit to experiments, it will be interesting to see if relaxations of them reveal new insights."

2.5 Reproducibility

I think the paper provided sufficiently enough theoretical details and derivation for anyone who has a background in kernel method and knowledgeable in statistical physics could follow the derivation. Also the paper provides enough pointers for motivated readers to learn about the background.

However on the empirical front, the provided details does not seem to be sufficient that at least the reviewer, myself, is not confident that enough details are provided to reproduce the result easily.

For example, experiments in Fig 5, what was the task that's learning on, learning rate, l2 regularization, parameterization, initialization weight/bias variances? It would benefit the readers to either 1) provide details of experiments in detail and/or 2) share code for reproducing experiments to understand proposed methods/analysis more carefully.

Thank you for stressing the lack of sufficient details for experimental procedures. We have substantially improved our manuscript in this regard. Because the changes are many, we will only list them here, and not quote the changes. We apologize in advance for the inconvenience.

- We revised experiment details section "SI.5 Experiment Details" in Supplementary Information for extended description of all procedures for experiments.
- All figure captions are revised with extended information, including tasks considered.
- We expanded our methods section "Diagonalizing the kernel on real datasets" to clarify our procedure in applying our method to real datasets.
- Most importantly, all the code to reproduce the experimental curves are now available in the GitHub Repository <https://github.com/Pehlevan-Group/kernel-generalization>.

About the reviewer’s specific question regarding Figure 6 (Figure 5 in the old manuscript): For all synthetic tasks with rotationally invariant kernels, the learning task is a target of the form $\bar{f}(\mathbf{x}) = c_k Q_k^{(D-1)}(\boldsymbol{\beta} \cdot \mathbf{x})$, where c_k is a constant, $\boldsymbol{\beta}$ is a random vector, and $Q_k^{(D-1)}$ is the k -th Gegenbauer polynomial. This kind of target functions (we call pure target) have a single mode k of spherical harmonics Y_{km} , hence in the E_g expression all target weights w_l vanish except $l = k$ (detailed in depth at the new section "SI.5.2 Synthetic Data Experiments on Unit Sphere"). We finally generate the training and test sets by drawing random inputs on unit (D-1)-sphere, and calculating the labels using the target function $\bar{f}(\mathbf{x})$. In the particular example presented in the new Fig.6 (old Fig.5), we picked $k = 1$ linear target. Other figures, e.g. Fig.5 (old Fig.4) contain nonlinear targets. For the neural network experiments, we initialized a single hidden layer neural network using NTK initialization scheme [24] with unit weight variance and zero bias variance. With the methods described above, we similarly generated our dataset and performed the training using ADAM optimizer with learning rate 0.008. Throughout the process, we used NeuralTangents package [32]. The code to reproduce the experimental curves are available in <https://github.com/Pehlevan-Group/kernel-generalization>.

2.6 Nits

We greatly appreciate for pointing out the mistakes/typos in the text.

- *L7 end: “new a theory” -> “a new theory”*

Thank you for pointing out this mistake. Typo has been fixed.

- *Fig2 caption: noise variance $\sigma = 0, 1.5$ (5 is missing)*

Thank you for pointing out this mistake. Typo has been fixed.

- *L47: consider replacing [10] and instead adding Matthews et al., “Gaussian Process Behaviour in Wide Deep Neural Networks”, ICLR 2018. While [10] is an important reference and requires citation but probably not here.*

We thank the reviewer for pointing this out. We added the reference to Matthews et al., but kept the reference to (Cho and Saul, 2009), which we believe is relevant.

- *Fig 4: (A-D) is missing in the actual figures*

Thank you for pointing out this mistake. Typo has been fixed.

- *Section 3.2.1: For spherical dataset on NTK consider also citing: Yang & Salman, "A Fine-Grained Spectral Perspective on Neural Networks" [49], 2019 and Bietti & Mairal, "On the Inductive Bias of Neural Tangent Kernels", NeurIPS 2019 [36].*

Thank you for your suggestions. The two papers the reviewer mentioned are indeed important references studying the inductive biases/implicit regularization of kernel machines and neural networks. Following the reviewer’s comment, we added the references in the 3rd paragraph of section "3.2.1: Dot Product Kernels, NTK and Wide Neural Networks" when discussing the implicit regularization of neural networks in line 370

“Neural networks are thought to generalize well because of implicit regularization [50, 36, 49]. This can be addressed with our formalism. For spherical data, we see that the implicit regularization of a neural network for each mode l is given by $\tilde{\lambda}_l = \frac{\sum_{k>l} \tilde{\eta}^k}{\tilde{\eta}^l}$. As an example, we calculate the spectrum of NTK for rectifier activations, and observe that the spectrum whitens with increasing depth [49], corresponding to larger $\tilde{\lambda}_l$ and therefore more regularization for small learning stages l (Fig. 6B)”

2.7 Recommendation

I believe this submission is a very solid work tackling an important problem in machine learning applying tools from theoretical physics. In general technique used is concrete and the results are convincing and illuminates various phenomenologies. There are some concerns on significance and novelty that should be clearly addressed by the authors.

We are gratified by the reviewer’s recommendation. We hope our revision addresses the concerns on significance and novelty.

3 Reviewer #3 (Remarks to the Author)

3.1 Contribution

The authors of the paper present new insights on the generalization error of kernel methods. Using tools from physics, they investigate this generalization, obtaining a general formula which matches empirically the generalization error of kernel regression, even for finite number of data points. More precisely, using the fact that the minimization problem can be approximated by a statistical model and letting the temperature going to 0, the problem becomes a computation of a difficult integral. This computation is done using a Gaussian approximation, the so-called replica method and the method of steepest descent. This leads the author to a simple (yet interesting) expression for the generalization error of kernel methods. In this general framework, they find that components of the target function along the eigenfunctions of the Kernel with large eigenvalues are “fitted faster” as the number of parameters increases: this is the spectral bias property. Using this insight, they are able to quantify the difficulty of a task given a kernel. All these assertions are backed by real world experiments using the MNIST and CIFAR dataset. Then, they apply their main formula to study the asymptotic behavior of the generalization error E_g to two settings:

1. Band Limited Kernels: these are toy models in order to study the more general setting of Rotation Invariant Kernel. When the number of data points $P \rightarrow \infty$ and the number of constant eigenvalues $N \rightarrow \infty$ with a constant ratio $\alpha = P/N$, the limiting E_g can be computed explicitly. They can thus study its behavior as a function of α , of the ridge λ and the amount of noise σ^2 . It

has to be noticed that if they recover the so-called double descent, they obtain a stronger result since they manage to compute explicitly the $\lambda - \sigma^2$ phase diagram for the double descent, and the different asymptotic decays of E_g in α .

2. *Rotation Invariant Kernels: one example of such Rotation Invariant Kernel, which is studied empirically and theoretically in this paper, is the recent NTK for infinitely large Neural Networks. In the spectrum of Rotation Invariant Kernels, eigenvalues have multiplicities which grow as the dimension of the data D goes to infinity at different speeds. This lead the authors to define different “learning episodes” which are obtained using different asymptotics for P and D . They note that indeed such limit can be described in term of learning episodes in the sense that larger eigenvalues are already learned and do not appear in the generalization error, and only the components along the eigenfunctions selected by the asymptotics of P and D are learned as the number of parameters increase. They also give interesting insight on the role of the smaller eigenvalues of the kernel and of the residue of the target function on the corresponding eigenfunctions during this learning episode: the eigenvalues act as regularizer as they induce an effective regularization whereas the residue of the function act as a noise term. This leads to a double descent curve during this learning episode, which leads ultimately to a multiple descent curve when considering all stages.*

We thank the reviewer for a careful reading of our work. We address his/her comments in detail below. We are grateful for the constructive suggestions.

3.2 Impression

The study of generalization of kernel methods using the statistical tool box is very interesting and the key results are well illustrated by real world experiments. What is interesting in this paper is that it provides a general formula for general kernel but also, most importantly, a lot of insights and new results about the specific but important kernels which are the Rotation Invariant Kernel. It was very appreciated to have the simpler section about Band Limited Kernels since most of the intuitions are already convey in this section. The concepts, results and computational techniques are well introduced and the computations are, in my opinion, sufficiently detailed. Overall, I think the article is very interesting and very well written, reaches the level of publications in Nature Communications and therefore worthy of publication after appropriate corrections and modifications by the authors.

We are gratified to read the reviewer’s encouraging impressions. We thank again for your constructive suggestions and comments.

3.3 Comments

What follows are minor comments, only on the main text.

- *Correct the λ line 134 in Equation (5): it should be η .*

We fixed the typo in line 134 (new line 144) and thank the reviewer for pointing it out.

- *In the paper “Kernel Alignment Risk Estimator: Risk Prediction from Training Data” [NeurIPS 2020][51] which does not study as deeply the asymptotic of the generalization error as in this paper, a similar equation as Equation (4) is proven mathematically. The κ plays the role of the Signal Capture Threshold ϑ and the $\frac{1}{1-\gamma}$ should be linked to its derivative wrt to the ridge $\partial_\lambda \vartheta$. It would be interesting to discuss the similarities and difference between these two formulas for the E_g and more generally between these two articles.*

We thank the reviewer for bringing up this very relevant paper. Indeed, we should have provided a detailed discussion of it. In the current version, we introduced a section in SI to discuss the relation between the two works, which we refer to from the main text.

While there are similarities between theories, they are different. In terms of similarities, both theories predict risk which is invariant under the simultaneous rescaling $K \rightarrow \alpha K, \lambda \rightarrow \alpha \lambda$. As the reviewer points out, our κ plays the role of the signal capture ratio of KARE and its derivative with respect to the ridge λ does in fact produce the prefactor $\partial_\lambda \kappa = \frac{1}{1-\gamma}$ which arises in our theory. Further, as we now provide in Equation SI.59, we predict that the training error is proportional to generalization error with proportionality constant $\frac{\lambda^2}{\kappa^2}$, as predicted also by KARE. However, upon close inspection our theory and KARE are not identical. This can be seen most easily in the ridgeless $\lambda \rightarrow 0$ limit. In this case, the KARE gives (in our notation)

$$E_{KARE}(P) = \frac{P^{-\beta}}{\left(\sum_\rho \eta_\rho^{-1}\right)^2} \sum_\rho \frac{\bar{w}_\rho^2}{\eta_\rho} \quad (\text{R.7})$$

for some constant β which only depends on how entries of the kernel were scaled with P . In this limit, all mode errors scale with P in an identical manner. However, our theory predicts the following generalization error

$$E_g = \frac{\kappa^2}{1-\gamma} \sum_\rho \frac{\bar{w}_\rho^2 \eta_\rho}{(\eta_\rho P + \kappa)^2}, \quad \sum_\rho \frac{\eta_\rho}{\eta_\rho P + \kappa} = 1, \quad \gamma = P \sum_\rho \frac{\eta_\rho^2}{(\eta_\rho P + \kappa)^2} \quad (\text{R.8})$$

Studying an approximation of our formula where $\eta_\rho \sim \rho^{-b}$ and $\bar{w}_\rho^2 \eta_\rho \sim \rho^{-a}$, we find that $E_g \sim P^{-\min\{a-1, 2b\}}$ when $\lambda \rightarrow 0$. The exponent for this power law depends on the spectrum and the task, indicating that our theory predicts other possibilities than the E_{KARE} scaling in the $\lambda \rightarrow 0$ limit. We added a paragraph to the related work section comparing these two theories and mentioning these differences.

The reference in the main text on line 390 reads:

"We also note a closely related recent study [51] that utilizes random matrix theory to study generalization in kernel regression, which we discuss in detail in SI."

The supplementary discussion reads:

"SI.7.1 Relation to Kernel Alignment Risk Estimator (KARE)

A closely related study to our theory on the generalization of kernel methods utilizes random matrix theory to arrive at a related, but different, generalization prediction [51]. Their Kernel Alignment Risk Estimator (KARE) was shown to accurately predict generalization on the Higgs and MNIST datasets. The *signal capture threshold* (SCT) which arises in KARE theory is equivalent to the quantity we denote as κ , which can be interpreted as the resolvent of a generalized Wishart matrix (SI.3.1). The mode independent prefactor in our theory $\frac{1}{1-\gamma}$, which plays an important role in the KARE theory, can be obtained from differentiation κ with respect to the ridge $\partial_\lambda \kappa$. Despite these similarities, our theory is not equivalent to KARE, which can be easily seen in the $\lambda \rightarrow 0$ limit, in which KARE scales with P in a manner that does not depend on the kernel or task spectra whereas our learning curves still depend on η_ρ and \bar{w}_ρ^2 even in the ridgeless limit. Our theory works well even in the $\lambda = 0$ simulations, for example in Figure 3A and 4B.

KARE may appear easier to implement than the theory we present here, since its only operations involve taking inverses, traces and quadratic forms of kernel gram matrices, whereas our theory involves a full eigendecomposition. In terms of time complexity, however, both methods scale as $O(M^3)$ for a sampled dataset of size M ."

- *Also in your experiments, in order to use the formula (4), one has to approximate the eigenvalues of the continuous kernel using the training data, besides one has to also estimates the weights of the target label function. One can avoid this by considering the “Kernel Alignment Risk Estimator” (see previous point).*

Thank you again for mentioning this paper which is quite relevant to the type of discussion we carry out. We interpret the reviewer’s comment as pointing to a computational complexity advantage of KARE over our estimator. We respectfully disagree.

The reviewer correctly points out the convenience of KARE’s implementation, which does not require any eigendecompositions, and rather only uses quadratic forms, inverses, and traces of the original kernel gram matrix whereas our theory requires computing eigenvectors Φ and eigenvalues Λ . However, inversion of the $M \times M$ gram matrix in KARE still costs $O(M^3)$ which has the same asymptotic time complexity as full eigendecomposition. We discuss this in our Supplementary Discussion, quoted above.

Further, our theory can be implemented without explicitly calculating target weights by using a procedure as in KARE. The numerator of KARE which involves labels can be written as

$$\mathbf{y}^\top (\mathbf{K}/P + \lambda \mathbf{I})^{-2} \mathbf{y} \tag{R.9}$$

while in our theory, the generalization error can be written as

$$\frac{(1 - \gamma)}{\kappa^2} E_g = \mathbf{y}^\top \Phi (\Lambda P + \kappa \mathbf{I})^{-2} \Phi^\top \mathbf{y} = \mathbf{y}^\top (\mathbf{K}P + \kappa \mathbf{I})^{-2} \mathbf{y}. \tag{R.10}$$

The expression on the right reveals that, if we have computed κ for the values of P we are interested in, then the generalization error can be computed with inversion of the matrix $\mathbf{K}P + \kappa \mathbf{I}$, which does not require explicitly computing or storing Φ , or projecting labels on them to calculate target weights. In summary, compared to KARE, we need to do the extra calculation of the eigenspectrum of the kernel in order to determine κ and γ , but this does not increase the overall complexity of the algorithm.

- *Since you are talking about Task-Model Alignment, could you elaborate on the link between your definition and results and the Kernel Alignment as defined for example in [52] <http://papers.neurips.cc/paper/1946-on-kernel-target-alignment.pdf> ?*

Thank you for bringing up this paper to our attention, which we should not have missed. The paper "On Kernel-Target Alignment" defines a particular metric for alignment by the angle/inner-product between two bi-dimensional kernels with respect to the data distribution and derives concentration bounds on it to infer generalization bounds for a binary classification problem. We focus on regression, and derive analytical expressions for the typical case that is predictive on real data and only measure alignment through cumulative power $C(\rho)$ defined in main text.

On the other hand, we can show that their predictions about the optimally aligned kernel for a given task is consistent with our theory. The prediction of [52] states that an optimal kernel

for a given task $y(x)$ is $K(x, x') = y(x)y(x')$. Note that this is also a Mercer decomposition, ($\int dx p(x)K(x, x')y(x) = \lambda y(x')$, where $\lambda = \int dx p(x)y(x)^2$). This optimal kernel is one for which all the weights are placed into the first eigenfunction. By our "task-model alignment" principle, $y(x)$ would be the target function whose generalization error falls fastest under this kernel, because $C(1) = 1$ already. Further, the corresponding generalization error would have the same form as the band-limited case we studied with $N = 1$. A similar comment has also been made in [49] where they pick their target function to be one of the eigenfunctions of the kernel.

We expanded our discussion to cover this link suggested by the reviewer under the Supplementary Discussion:

"SI.7.2 Relation to Kernel Alignment

In this paper, we introduced the idea of task-model alignment. A similar notion of kernel compatibility with target was discussed in [52, 53] (latter of which introduces algorithms to learn better aligned kernels), by introducing an "alignment" metric defined by:

$$A(K_1(x, x'), K_2(x, x')) = \frac{\langle K_1(x, x'), K_2(x, x') \rangle_p}{\sqrt{\langle K_1(x, x'), K_1(x, x') \rangle_p \langle K_2(x, x'), K_2(x, x') \rangle_p}}, \quad (\text{SI.142})$$

where $\langle k_1(x, x'), k_2(x, x') \rangle_p = \int dp(x)dp(x')k_1(x, x')k_2(x, x')$ denotes the inner product of two bi-dimensional functions with respect to the data distribution $p(x)$. The so-called "kernel-target alignment" between a kernel $K(x, x')$ and target function $\bar{f}(x)$ is then defined by $A(K(x, x'), \bar{f}(x)\bar{f}(x'))$.

To get further insight about the differences between the kernel alignment metric and our task-model alignment, we can evaluate the kernel alignment metric using the eigendecompositions of the kernel and the target function. We obtain:

$$A(K(\mathbf{x}, \mathbf{x}'), \bar{f}(\mathbf{x})\bar{f}(\mathbf{x}')) = \frac{1}{\sum_{\rho} \eta_{\rho} \bar{w}_{\rho}^2} \frac{\sum_{\rho} \eta_{\rho}^2 \bar{w}_{\rho}^2}{\sqrt{\sum_{\rho} \eta_{\rho}^2}}, \quad (\text{SI.143})$$

which is a scalar between $[0, 1]$. In contrast, the task-model alignment states that when the cumulative power $C(\rho) = \frac{\sum_{\rho' \leq \rho} \eta_{\rho'} \bar{w}_{\rho'}^2}{\sum_{\rho'} \eta_{\rho'} \bar{w}_{\rho'}^2}$ at each mode ρ for a target functions is entry-wise larger than the $C(\rho)$ for another target function, the target function with larger $C(\rho)$ is learned with less training data then the other. While the cumulative power contains information about each learning stage ρ , kernel-alignment is an aggregate measure of how kernel and task is aligned.

To study the similarities between our prediction and [52] in a simple context, let us consider the regression task with the kernel $K(x, x') = \bar{f}(x)\bar{f}(x')$ and target $\bar{f}(x)$. This kernel and the target are perfectly aligned under the metric of [52] ($A=1$). The kernel has only a single eigenfunction $\phi(x) = \bar{f}(x)/\sqrt{\langle \bar{f}^2(x) \rangle_p}$ and hence a single non-zero eigenvalue $\eta = \langle \bar{f}^2(x) \rangle_p$. Furthermore, the eigenspace expansion of the target function gives trivially $\bar{w} = 1$. Hence all the target power is placed in the first eigenfunction. Our theory implies that the \bar{f} is the target function whose generalization error falls fastest under this kernel ($C(1) = 1$). Interestingly, in this case the generalization error has the same form as the band-limited case. "

- *Correct Figure 3, the labels for the x should be α and not P . Also it not clear at all what is the horizontal line in the phase diagram (I understood it was linked with the plot C but it took me some time), it would be better actually to remove this line and to keep only the dots and to explain what are these dots.*

We thank the reviewer for advice on Figure 3. The new version is updated with the correct x axis label. As per the suggestions of the reviewer, the horizontal dashed line has been removed and we explain the dots in the figure caption.

On the phase diagram, the colored dots indicate the parameters of the experiment curves with the same color presented in panel C. We try to make the point that when the noise is fixed, there is an optimal regularizer for each noise level (indicated with yellow dashed line in the phase diagram) which yields the best generalization performance.

- *Using K both for the kernel and as an index for the eigenvalues might not be a good idea.*

We tried to always use capitalization when referring to the kernel K as a function, but used k occasionally to index modes.

- *In Figure 4, labels A, B, C, D are missing.*

Thanks. We fixed the labels for Figure 5 (previously Figure 4).

- *Same figure, the three other figures, is it $E_g^{(1)}$ or E_g which is drawn ?(since there are multiple descent, this would be in contradiction with the main text if it was $E_g^{(1)}$). Besides, it is slightly confusing that you use both P and $P/N(D, 1)$ knowing that in fact $N(D, 1)$ is fixed. Would it not be clearer to use the same x axis for these three pictures ?*

Thank you very much for bringing up this mistake. It should have been E_g . In our notation, $E_g^{(1)}$ denotes the asymptotic generalization error corresponding to first learning stage and indeed there is only a single peak in each learning stage. We updated the figure (new Figure 5) to address this mistake.

Also, as suggested, we changed the x axis of Figure 5B to $P/N(D, 1)$ to avoid confusion.

References

- [1] Tengyuan Liang, Alexander Rakhlin, and Xiyu Zhai. On the multiple descent of minimum-norm interpolants and restricted lower isometry of kernels. In Jacob Abernethy and Shivani Agarwal, editors, *Proceedings of Thirty Third Conference on Learning Theory*, volume 125 of *Proceedings of Machine Learning Research*, pages 2683–2711. PMLR, 09–12 Jul 2020.
- [2] Preetum Nakkiran, Prayaag Venkat, Sham M. Kakade, and Tengyu Ma. Optimal regularization can mitigate double descent. In *International Conference on Learning Representations*, 2021.
- [3] Stéphane d’Ascoli, Levent Sagun, and Giulio Biroli. Triple descent and the two kinds of overfitting: Where and why do they appear? *Advances in Neural Information Processing Systems*, 2020.
- [4] Lin Chen, Yifei Min, Mikhail Belkin, and amin karbasi. Multiple descent: Design your own generalization curve, 2021.

- [5] Ben Adlam and Jeffrey Pennington. The neural tangent kernel in high dimensions: Triple descent and a multi-scale theory of generalization. In *International Conference on Machine Learning*, pages 74–84. PMLR, 2020.
- [6] Zhenyu Liao, Romain Couillet, and Michael W. Mahoney. A random matrix analysis of random fourier features: beyond the gaussian kernel, a precise phase transition, and the corresponding double descent. In *Advances in Neural Information Processing Systems*. MIT Press, 2020.
- [7] Tengyuan Liang, Alexander Rakhlin, and Xiyu Zhai. On the multiple descent of minimum-norm interpolants and restricted lower isometry of kernels. volume 125 of *Proceedings of Machine Learning Research*, pages 2683–2711. PMLR, 09–12 Jul 2020.
- [8] Stefano Spigler, Mario Geiger, and Matthieu Wyart. Asymptotic learning curves of kernel methods: empirical data versus teacher–student paradigm. *Journal of Statistical Mechanics: Theory and Experiment*, 2020(12):124001, dec 2020.
- [9] Song Mei and Andrea Montanari. The generalization error of random features regression: Precise asymptotics and double descent curve. *arXiv preprint arXiv:1908.05355*, 2019.
- [10] Stéphane d’Ascoli, Maria Refinetti, Giulio Biroli, and Florent Krzakala. Double trouble in double descent : Bias and variance(s) in the lazy regime. In *Proceedings of the 37th International Conference on Machine Learning*, 2020.
- [11] Preetum Nakkiran. More data can hurt for linear regression: Sample-wise double descent. *arXiv preprint arXiv:1912.07242*, 2019.
- [12] Mikhail Belkin, Daniel Hsu, Siyuan Ma, and Soumik Mandal. Reconciling modern machine-learning practice and the classical bias–variance trade-off. *Proceedings of the National Academy of Sciences*, 116(32):15849–15854, 2019.
- [13] Preetum Nakkiran, Gal Kaplun, Yamini Bansal, Tristan Yang, Boaz Barak, and Ilya Sutskever. Deep double descent: Where bigger models and more data hurt. In *International Conference on Learning Representations*, 2019.
- [14] Arthur Jacot, Berfin Simsek, Francesco Spadaro, Clement Hongler, and Franck Gabriel. Implicit regularization of random feature models. In Hal Daumé III and Aarti Singh, editors, *Proceedings of the 37th International Conference on Machine Learning*, volume 119 of *Proceedings of Machine Learning Research*, pages 4631–4640. PMLR, 13–18 Jul 2020.
- [15] Trevor Hastie, Andrea Montanari, Saharon Rosset, and Ryan J. Tibshirani. Surprises in high-dimensional ridgeless least squares interpolation. *arXiv preprint arXiv:1903.08560*, 2019.
- [16] S Spigler, M Geiger, S d’Ascoli, L Sagun, G Biroli, and M Wyart. A jamming transition from under-to over-parametrization affects generalization in deep learning. *Journal of Physics A: Mathematical and Theoretical*, 52(47):474001, 2019.
- [17] Federica Gerace, Bruno Loureiro, Florent Krzakala, Marc Mezard, and Lenka Zdeborova. Generalisation error in learning with random features and the hidden manifold model. In Hal Daumé III and Aarti Singh, editors, *Proceedings of the 37th International Conference on Machine Learning*, volume 119 of *Proceedings of Machine Learning Research*, pages 3452–3462. PMLR, 13–18 Jul 2020.

- [18] A Krogh and J. Hertz. Generalization in a linear perceptron in the presence of noise. *Journal of Physics A: Mathematical and General*, 25:1135, 01 1999.
- [19] J A Hertz, A Krogh, and G I Thorbergsson. Phase transitions in simple learning. *Journal of Physics A: Mathematical and General*, 22(12):2133–2150, jun 1989.
- [20] Marc G Genton. Classes of kernels for machine learning: a statistics perspective. *Journal of machine learning research*, 2(Dec):299–312, 2001.
- [21] Carl Edward Rasmussen and Christopher K. I. Williams. *Gaussian Processes for Machine Learning (Adaptive Computation and Machine Learning)*. The MIT Press, 2005.
- [22] Mario Geiger, Arthur Jacot, Stefano Spigler, Franck Gabriel, Levent Sagun, Stéphane d’Ascoli, Giulio Biroli, Clément Hongler, and Matthieu Wyart. Scaling description of generalization with number of parameters in deep learning. *Journal of Statistical Mechanics: Theory and Experiment*, 2020(2):023401, Feb 2020.
- [23] Youngmin Cho and Lawrence K Saul. Kernel methods for deep learning. In *Advances in neural information processing systems*, pages 342–350, 2009.
- [24] Arthur Jacot, Franck Gabriel, and Clément Hongler. Neural tangent kernel: Convergence and generalization in neural networks. In *Advances in neural information processing systems*, pages 8571–8580, 2018.
- [25] Yann LeCun, Corinna Cortes, and CJ Burges. Mnist handwritten digit database. *ATT Labs [Online]*. Available: <http://yann.lecun.com/exdb/mnist>, 2, 2010.
- [26] Sina Alemohammad, Zichao Wang, Randall Balestriero, and Richard Baraniuk. The recurrent neural tangent kernel. In *International Conference on Learning Representations*, 2021.
- [27] Greg Yang. Tensor programs i: Wide feedforward or recurrent neural networks of any architecture are gaussian processes. In *NeurIPS 2019*, December 2019. ArXiv.
- [28] Greg Yang. Tensor programs ii: Neural tangent kernel for any architecture. *2006.14548*, 2020.
- [29] Greg Yang. Tensor programs iii: Neural matrix laws, 2020.
- [30] Stanislav Fort, Gintare Karolina Dziugaite, Mansheej Paul, Sepideh Kharaghani, Daniel M Roy, and Surya Ganguli. Deep learning versus kernel learning: an empirical study of loss landscape geometry and the time evolution of the neural tangent kernel. *arXiv preprint arXiv:2010.15110*, 2020.
- [31] Lénaïc Chizat and Francis Bach. Implicit bias of gradient descent for wide two-layer neural networks trained with the logistic loss. In Jacob Abernethy and Shivani Agarwal, editors, *Proceedings of Thirty Third Conference on Learning Theory*, volume 125 of *Proceedings of Machine Learning Research*, pages 1305–1338. PMLR, 09–12 Jul 2020.
- [32] Roman Novak, Lechao Xiao, Jiri Hron, Jaehoon Lee, Alexander A. Alemi, Jascha Sohl-Dickstein, and Samuel S. Schoenholz. Neural tangents: Fast and easy infinite neural networks in python. In *International Conference on Learning Representations*, 2020.
- [33] Sanjeev Arora, Simon S Du, Wei Hu, Zhiyuan Li, Russ R Salakhutdinov, and Ruosong Wang. On exact computation with an infinitely wide neural net. In *Advances in Neural Information Processing Systems*, pages 8139–8148, 2019.

- [34] Behnam Neyshabur, Ryota Tomioka, and Nathan Srebro. In search of the real inductive bias: On the role of implicit regularization in deep learning. *arXiv preprint arXiv:1412.6614*, 2014.
- [35] Jaehoon Lee, Lechao Xiao, Samuel Schoenholz, Yasaman Bahri, Roman Novak, Jascha Sohl-Dickstein, and Jeffrey Pennington. Wide neural networks of any depth evolve as linear models under gradient descent. In *Advances in neural information processing systems*, pages 8570–8581, 2019.
- [36] Alberto Bietti and Julien Mairal. On the inductive bias of neural tangent kernels. In H. Wallach, H. Larochelle, A. Beygelzimer, F. d'Alché-Buc, E. Fox, and R. Garnett, editors, *Advances in Neural Information Processing Systems*, volume 32, pages 12893–12904. Curran Associates, Inc., 2019.
- [37] Behrooz Ghorbani, Song Mei, Theodor Misiakiewicz, and Andrea Montanari. Linearized two-layers neural networks in high dimension. *arXiv preprint arXiv:1904.12191*, 2019.
- [38] Daniel Soudry, Elad Hoffer, and Nathan Srebro. The implicit bias of gradient descent on separable data. *Journal of Machine Learning Research*, 19, 10 2017.
- [39] Rainer Dietrich, Manfred Opper, and Haim Sompolinsky. Statistical mechanics of support vector networks. *Phys. Rev. Lett.*, 82:2975–2978, Apr 1999.
- [40] M. Opper and R. Urbanczik. Universal learning curves of support vector machines. *Phys. Rev. Lett.*, 86:4410–4413, May 2001.
- [41] Madhu Advani and Surya Ganguli. Statistical mechanics of optimal convex inference in high dimensions. *Phys. Rev. X*, 6:031034, Aug 2016.
- [42] Peter Sollich. Finite-size effects in learning and generalization in linear perceptrons. *Journal of Physics A: Mathematical and General*, 27(23):7771, 1994.
- [43] Peter Sollich. Learning curves for gaussian processes. *Advances in neural information processing systems*, pages 344–350, 1999.
- [44] Peter Sollich and Anason Halees. Learning curves for gaussian process regression: Approximations and bounds. *Neural computation*, 14(6):1393–1428, 2002.
- [45] Blake Bordelon, Abdulkadir Canatar, and Cengiz Pehlevan. Spectrum dependent learning curves in kernel regression and wide neural networks. In *Proceedings of the 37th International Conference on Machine Learning*, 2020.
- [46] Michael Peskin and Dan Schroeder. *An Introduction to Quantum Field Theory*. 1995.
- [47] Omry Cohen, Or Malka, and Zohar Ringel. Learning curves for deep neural networks: a gaussian field theory perspective. *arXiv preprint arXiv:1906.05301*, 2019.
- [48] Bruno Loureiro, Cédric Gerbelot, Hugo Cui, Sebastian Goldt, Florent Krzakala, Marc Mézard, and Lenka Zdeborová. Capturing the learning curves of generic features maps for realistic data sets with a teacher-student model. *arXiv preprint arXiv:2102.08127*, 2021.
- [49] Greg Yang and Hadi Salman. A fine-grained spectral perspective on neural networks, 2020.
- [50] C Zhang, S Bengio, M Hardt, B Recht, and O Vinyals. Understanding deep learning requires rethinking generalization. In *5th Int. Conf. on Learning Representations (ICLR 2017)*, 2016.

- [51] Arthur Jacot, Berfin Şimşek, Francesco Spadaro, Clément Hongler, and Franck Gabriel. Kernel alignment risk estimator: Risk prediction from training data, 2020.
- [52] Nello Cristianini, John Shawe-Taylor, André Elisseeff, and Jaz Kandol a. On kernel-target alignment. In T. Dietterich, S. Becker, and Z. Ghahramani, editors, *Advances in Neural Information Processing Systems*, volume 14, pages 367–373. MIT Press, 2002.
- [53] Corinna Cortes, Mehryar Mohri, and Afshin Rostamizadeh. Algorithms for learning kernels based on centered alignment. *The Journal of Machine Learning Research*, 13(1):795–828, 2012.

Reviewer #1 (Remarks to the Author):

The authors did an excellent job at addressing my comments, through substantial editing of the text and the addition of many valuable new results (particularly the new results on the bias-variance decomposition).

I therefore fully recommend this article for publication.

Stéphane d'Ascoli

Reviewer #2 (Remarks to the Author):

I thank the authors for taking their time and effort to address all the issues and questions raised by the reviewers. Reading the extensive response and revisions, I believe the paper has significantly improved and addresses all the major concerns. I certainly learned a lot from insights from other reviewers and the author's thorough responses. Added supplement in relation to other methods (SI. 7) is very helpful and I am very thankful to the authors for writing out such careful comparisons.

The author's explanations on my concerns and questions are quite satisfactory and I'm happy to recommend acceptance of the paper. I believe this work will serve as a major toolbox of understanding generalization properties of overparameterized neural networks and kernels.

Reviewer #3 (Remarks to the Author):

I would like to thank the authors for their very exhaustive response. The various points were addressed in great detail and I am totally satisfied with the answers and the proposed changes. I therefore recommend the publication.

Response to reviewer comments on "Spectral Bias and Task-Model Alignment Explain Generalization in Kernel Regression and Infinitely Wide Neural Networks"

Abdulkadir Canatar, Blake Bordelon and Cengiz Pehlevan

We thank all reviewers for their encouraging, insightful, and constructive comments and suggesting our work for publication. Their comments led a substantial expansion of our work and made the main points of our paper stronger.

NOTE: The referee comments are set in *violet*, and our replies in black.

1 Reviewer #1

The authors did an excellent job at addressing my comments, through substantial editing of the text and the addition of many valuable new results (particularly the new results on the bias-variance decomposition). I therefore fully recommend this article for publication. Stéphane d'Ascoli

We are grateful for the reviewer's excellent questions and insights causing a significant improvement of our paper. We are thankful for recommending our work for publication.

2 Reviewer #2 (Remarks to the Author)

I thank the authors for taking their time and effort to address all the issues and questions raised by the reviewers. Reading the extensive response and revisions, I believe the paper has significantly improved and addresses all the major concerns. I certainly learned a lot from insights from other reviewers and the author's thorough responses. Added supplement in relation to other methods (SI. 7) is very helpful and I am very thankful to the authors for writing out such careful comparisons.

The author's explanations on my concerns and questions are quite satisfactory and I'm happy to recommend acceptance of the paper. I believe this work will serve as a major toolbox of understanding generalization properties of overparameterized neural networks and kernels.

The reviewer's detailed comments and suggestions have significantly improved our paper and we are thankful for such careful assessment. We hope that the publication of this work will contribute to the field of generalization in machine learning.

3 Reviewer #3 (Remarks to the Author)

I would like to thank the authors for their very exhaustive response. The various points were addressed in great detail and I am totally satisfied with the answers and the proposed changes. I therefore recommend the publication.

We are grateful for the reviewer's careful assessment of the work and insightful suggestions which improved our paper greatly. We thank him/her for recommending our paper for publication.